# Learning the Minimum Action Distance

**Lorenzo Steccanella** [1]    **Joshua B. Evans** [2]    **Özgür Şimşek** [2]    **Anders Jonsson** [1]

## Abstract

This paper presents a state representation framework for Markov decision processes (MDPs) that can be learned solely from state trajectories, requiring neither reward signals nor the actions executed by the agent. We propose learning the *minimum action distance* (MAD), defined as the minimum number of actions required to transition between states, as a fundamental metric that captures the underlying structure of an environment. The MAD naturally enables critical downstream tasks such as goal-conditioned reinforcement learning and reward shaping by providing a dense, geometrically meaningful measure of progress. Our self-supervised learning approach constructs an embedding space where the distances between embedded state pairs correspond to their MAD, accommodating both symmetric and asymmetric approximations. We evaluate the framework on a comprehensive suite of environments with known MAD values, encompassing both deterministic and stochastic transition dynamics, discrete and continuous state spaces, and environments with noisy observations. Empirical results show that the proposed approach learns MAD representations more efficiently than existing methods, produces more accurate estimates of the true MAD, and improves performance on downstream goal-reaching tasks.

## 1. Introduction

In reinforcement learning (Sutton & Barto, 1998), an agent learns useful behaviors through continuing interaction with its environment. By observing the outcomes of its actions, a reinforcement learning agent learns over time how to select actions in a way that maximizes the expected cumulative reward it receives from its environment. A central challenge in reinforcement learning is giving agents the ability to generalize. An agent should be able to adapt quickly not only to previously unseen states, but also to variations of its environment that it has not previously observed.

One way to support this kind of generalization is to learn a metric that captures the similarity between two states in the environment. A useful state similarity metric can, for example, allow experience gathered in one region of the environment to inform behavior in another, support abstractions that group together equivalent or similar states, and enable transfer when the agent's task or the environment itself changes while preserving some of its underlying structure. Such metrics can also serve as heuristics in goal-conditioned reinforcement learning, where an agent must adapt quickly to achieve different goals in its environment.

The Minimum Action Distance (MAD) has proved useful as a state similarity metric, with impressive applications in several areas of reinforcement learning, including policy learning (Wang et al., 2023b; Park et al., 2023), reward shaping (Steccanella & Jonsson, 2022), and option discovery (Park et al., 2024a;b). While prior work has demonstrated the benefits of using the MAD, how best to approximate it remains an open problem. Existing methods have not been systematically evaluated on their ability to accurately approximate the MAD, and many rely on symmetric approximations, even though the true MAD is inherently asymmetric.

In this work, we propose an offline state representation learning framework that captures directed distances between states, without requiring action or reward information. By learning purely from sequences of state observations and focusing on global reachability rather than local dynamics, this approach is inherently robust to environmental stochasticity and remains agnostic to the data-collection policy. This allows for training on datasets of any quality, including purely random exploration, expert demonstrations, or fragmented trajectories that must be stitched together.

Figure 1 illustrates the steps of MAD representation learning: an agent collects state trajectories from an unknown environment, which are used to learn a state embedding that implicitly defines a distance function between states.

[1]Department of Information and Communication Technologies, Universitat Pompeu Fabra, Barcelona, Spain [2]Department of Computer Science, University of Bath, Bath, United Kingdom. Correspondence to: Lorenzo Steccanella <lorenzo.steccanella.w@gmail.com>.

*Proceedings of the 43rd International Conference on Machine Learning*, Seoul, South Korea. PMLR 306, 2026. Copyright 2026 by the author(s).

We make three main contributions towards fast, accurate approximation of the MAD. First, we propose two novel offline algorithms for learning MAD using only state trajectories collected by an agent interacting with its environment. Unlike previous work, the proposed algorithms naturally support both symmetric and asymmetric distances. Secondly, we define a novel quasimetric distance function that is computationally efficient and that, despite its simplicity, outperforms more elaborate quasimetrics in the existing literature. Finally, we introduce a diverse suite of environments — including those with discrete and continuous state spaces, stochastic and deterministic dynamics, and directed and undirected transitions — in which the ground-truth MAD is known, enabling a systematic and controlled evaluation of different MAD approximation methods.

## 2. Related Work

In applications such as goal-conditioned reinforcement learning (Ghosh et al., 2020) and stochastic shortest-path problems (Tarbouriech et al., 2021), the temporal distance is measured as the expected number of steps required to reach one state from another state under some policy. In contrast, the MAD is a lower bound on the number of steps based solely on the support of the transition function. This distinction makes the MAD efficient to compute and robust to changes in the transition probabilities as long as the support over next states remains the same, making it suitable for representation learning and transfer learning.

Prior work has explored the connection between the MAD and optimal goal-conditioned value functions (Kaelbling, 1993). Park et al. (2023) highlight this connection and propose a hierarchical approach that improves distance estimates over long horizons, and Park et al. (2024a) embed states into a learned latent space where the distance between embedded states directly reflects an on-policy measure of the temporal distance (Hartikainen et al., 2020). Park et al. (2024b) and Ma et al. (2022) extend this idea to the offline setting, learning embeddings from arbitrary experience such that Euclidean distances between state embeddings approximate the MAD. As an alternative to approximating the MAD using goal-conditioned value functions, Steccanella & Jonsson (2022) formulate learning a state embedding in which distances approximate the MAD as a constrained optimization problem, where bounds on the distance between embedded states are derived from state trajectory data. Although their formulations differ, these approaches ultimately seek to learn the same underlying quantity: the minimum number of actions required to move between two states.

These existing approaches share a common limitation: they rely on symmetric distance metrics such as the Euclidean distance between state embeddings to approximate the MAD. As such, they cannot capture the asymmetry of the true

MAD in environments with irreversible dynamics. In contrast, the approach we develop here supports the use of asymmetric distance metrics (or, *quasimetrics*), which can better capture the directional structure in many environments.

Some prior work has already explored the use of quasimetrics in reinforcement learning. Wang et al. (2023b) learn an asymmetric distance function that approximates the MAD by preserving local structure while maintaining global distances. Their method differs from the one we propose in two ways. First, their method does not leverage observed distances along state trajectories as supervision for the learning process. Secondly, they use the Interval Quasimetric Embedding (IQE) (Wang & Isola, 2022) to learn the distance function. Dadashi et al. (2021) and Agarwal et al. (2021) learn embeddings and define a pseudometric between states as the Euclidean distance between their embeddings. Unlike our work, they use objective functions inspired by bisimulation to learn both state and state-action embeddings.

Successor features (Dayan, 1993; Barreto et al., 2017), and time-contrastive representations (Eysenbach et al., 2022) have also been used to define notions of temporal distance. Myers et al. (2024) introduces time-contrastive successor features, defining a distance metric based on the difference between discounted future occupancies of state features learned via time-contrastive learning. While their metric satisfies the triangle inequality and naturally handles both stochasticity and asymmetry, the resulting distances reflect expected discounted state visitations under a specific behavior policy and lack an intuitive interpretation. In contrast, approaches that approximate the MAD are naturally interpretable as a lower bound on the number of actions needed to transition between two states.

Laplacian-based representation learning methods (Wu et al., 2019; Machado, 2019; Wang et al., 2021; 2023a) learn embeddings from the spectral structure of random walks over the transition graph, producing representations that reflect global connectivity in the state space. However, these methods are typically defined on a symmetrized transition operator or undirected Laplacian, and the induced geometry measures diffusion-based similarity rather than directed reachability. As a result, distances in these embeddings are fundamentally symmetric and do not correspond to the minimum number of actions required to move between two states, making them poorly suited to environments with irreversible or asymmetric dynamics.

## 3. Background

In this section, we introduce the notation and concepts used throughout the paper. Given a finite set $\mathcal{X}$, we use $\Delta(\mathcal{X}) = \{p \in \mathbb{R}^{\mathcal{X}} \mid \sum_x p_x = 1, p_x \geq 0 \, (\forall x)\}$ to denote the probability simplex (i.e. the set of all probability

*Figure 1.* Schematic overview of MAD representation learning. From left to right: (1) the hidden environment graph, (2) trajectories collected by an unknown policy, (3) the embedding function $\phi : S \to \mathbb{R}^2$ and (4) the resulting MAD embedding space in $\mathbb{R}^2$.

distributions over $\mathcal{X}$). A rectified linear unit (ReLU) is a function $\mathrm{relu} : \mathbb{R}^d \to \mathbb{R}^d_{\geq 0}$ defined on any vector $x \in \mathbb{R}^d$ as $\mathrm{relu}(x) = [\max(0, x_i)]_{i=1}^{d}$.

**Markov Decision Processes (MDPs).** An MDP (Bellman, 1957) is a tuple $\mathcal{M} = \langle \mathcal{S}, \mathcal{A}, \mathcal{R}, \mathcal{P}, \mathcal{D}, \gamma \rangle$, where $\mathcal{S}$ is the state space, $\mathcal{A}$ is the action space, $\mathcal{R} : \mathcal{S} \times \mathcal{A} \to \mathbb{R}$ is the reward function, $\mathcal{P} : \mathcal{S} \times \mathcal{A} \to \Delta(\mathcal{S})$ is the transition kernel, $\mathcal{D} \in \Delta(\mathcal{S})$ is the initial state distribution, and $\gamma \in [0, 1]$ is the discount factor. At each time $t$, the learning agent observes a state $s_t \in \mathcal{S}$, selects an action $a_t \in \mathcal{A}$, receives a reward $r_t = \mathcal{R}(s_t, a_t)$ and transitions to a new state $s_{t+1} \sim \mathcal{P}(s_t, a_t)$. The learning agent selects actions using a policy $\pi : \mathcal{S} \to \Delta(\mathcal{A})$, a mapping from states to probability distributions over actions. In our work, the state space $\mathcal{S}$ can be either discrete or continuous.

We work in the standard *discrete-time* MDP setting, where each transition corresponds to exactly one decision step and therefore has unit cost. Consequently, the Minimum Action Distance represents the minimum number of actions needed to move between states. This interpretation is natural for the benchmarks considered in this paper, even when the underlying state space is continuous, because the agent still interacts with the environment at discrete decision times.

The proposed framework could also be extended beyond this setting. In a semi-Markov decision process (SMDP) (Sutton et al., 1999), an action may persist for a variable duration, so different transitions need not consume the same amount of time. In a continuous-time control problem, time evolves continuously rather than in unit decision steps. In either case, one would replace the unit-cost notion used here with a duration or elapsed-time cost associated with each transition. We do not consider these extensions in this paper, but making the distinction explicit is important because our notion of MAD is defined with respect to discrete decision steps, not physical time.

**Reinforcement learning (RL).** RL (Sutton & Barto, 2018) algorithms aim to learn a policy $\pi$ that maximizes some measure of expected future reward. In this paper, we consider the problem of representation learning, and so are not directly concerned with the problem of learning a policy.

Concretely, we wish to learn a distance function between pairs of states that can later be used by an RL agent to learn more efficiently. In this setting, we assume that the learning agent uses a behavior policy $\pi_b$ to collect trajectories. Since we are interested in learning a distance function over state pairs, actions are relevant only for determining possible transitions between states, and rewards are not relevant. Hence, for our purposes, a trajectory $\tau = (s_0, s_1, \ldots, s_n)$ is simply a sequence of states.

## 4. The Minimum Action Distance

Given an MDP $\mathcal{M} = \langle \mathcal{S}, \mathcal{A}, \mathcal{R}, \mathcal{P}, \mathcal{D}, \gamma \rangle$ and a state pair $(s, s') \in \mathcal{S}^2$, the Minimum Action Distance, $d_{\mathrm{MAD}}(s, s')$, is defined as the minimum number of decision steps needed to transition from $s$ to $s'$.

**Definition 4.1** (Minimum Action Distance). For a fixed policy $\pi$, define the minimum hitting time from state $s$ to state $s'$ as

$$T_{s \to s'}^{\pi} = \min \{t \in \mathbb{N} \cup \{\infty\} : \mathbb{P}(S_t = s' \mid S_0 = s, \pi) > 0\}.$$

That is, $T_{s \to s'}^{\pi}$ denotes the earliest decision step at which state $s'$ is reachable from $s$ with non-zero probability under policy $\pi$.

The Minimum Action Distance is defined as

$$d_{\mathrm{MAD}}(s, s') = \inf_{\pi} T_{s \to s'}^{\pi}.$$

In deterministic MDPs, the MAD is always realizable using an appropriate policy; in stochastic MDPs, the MAD is a lower bound on the actual number of decision steps required by any policy. This makes the MAD robust to environmental stochasticity, as its value depends only on the support of the transition kernel $\mathcal{P}$.

Let $R \subseteq \mathcal{S} \times \mathcal{S}$ be the one-step reachability relation defined by $(s, s') \in R \iff \exists a \in \mathcal{A}$ such that $\mathcal{P}(s' \mid s, a) > 0$. That is, $R$ contains all state pairs $(s, s')$ such that $s'$ is reachable from $s$ in a single decision step. In a standard discrete-time MDP, each transition represents an atomic decision step, which we treat as having a unit cost of 1.

While we focus on discrete-time MDPs where each action takes exactly one step, this framework could be extended to Semi-Markov Decision Processes (SMDPs) or continuous-time systems by replacing the unit cost with a duration function $\Delta t(s, a)$.

We can characterize $d_{\mathrm{MAD}}$ as the unique solution to the following constrained optimization problem:

$$d_{\mathrm{MAD}} = \arg\max_{d} \sum_{(s,s')\in\mathcal{S}^2} d(s, s'), \tag{1}$$

$$\text{s.t.} \quad d(s, s) = 0 \quad \forall s \in \mathcal{S}, \tag{Identity}$$

$$d(s, s') \leq 1 \quad \forall (s, s') \in R, \tag{One-Step}$$

$$d(s, s') \leq d(s, s'') + d(s'', s') \quad \forall s, s', s'' \in \mathcal{S}. \tag{Triangle Ineq.}$$

The *One-Step* constraint enforces that the distance between states reachable in a single action is at most 1. Because the objective is to maximize the sum of all distances, the optimal solution $d^*$ will push these values to the highest possible value allowed by the constraints. Specifically, for any $(s, s') \in R$ where $s \neq s'$, the optimal distance is exactly $d^*(s, s') = 1$. The *Triangle Inequality* then ensures these unit costs are propagated globally, such that $d^*(s, s')$ matches the length of the shortest sequence of actions connecting the two states.

**Theorem 4.2** (Pointwise Maximality). *The Minimum Action Distance $d_{MAD}$ is the unique pointwise maximal function satisfying the constraints in equation 1. That is, for any feasible distance function $d$, $d(s, s') \leq d_{MAD}(s, s')$ for all $(s, s') \in \mathcal{S}^2$.*

The proof (see Appendix A) relies on the fact that any path of length $k$ implies $d(s, s') \leq k$ via the triangle inequality; since $d_{\mathrm{MAD}}$ is defined as the minimum such $k$, it represents the maximum possible value any consistent distance function can take.

If the state space $\mathcal{S}$ is finite, the constrained optimization problem is precisely the linear programming formulation of the all-pairs shortest-path problem for the directed graph $(\mathcal{S}, R)$ with unit edge costs. This graph can be viewed as a determinization of the MDP $\mathcal{M}$ (Yoon et al., 2007). In this case, $d_{\mathrm{MAD}}$ can be computed exactly using the Floyd–Warshall algorithm (Floyd, 1962; Warshall, 1962).

If the state space $\mathcal{S}$ is continuous, the relation $R$ remains well-defined, and the optimization problem still has a well-defined solution. However, the states can no longer be enumerated explicitly, and solving equation 1 directly becomes computationally infeasible. Moreover, the triangle inequality constraint introduces a number of inequalities that grows cubically with the size of the state space, making the formulation unsuitable as a practical learning objective.

To obtain a tractable formulation, we replace the search over arbitrary functions $d : \mathcal{S} \times \mathcal{S} \to \mathbb{R}_{\geq 0}$ with a parametrized family of distance functions defined through a learned state representation.

Specifically, we introduce a state embedding function $\phi_\theta : \mathcal{S} \to \mathbb{R}^k$ parametrized by $\theta$, and define the distance between two states as $d_\theta(s, s') = d_q\big(\phi_\theta(s), \phi_\theta(s')\big)$, where $d_q$ is a quasimetric on $\mathbb{R}^k$ (see Appendix C).

A quasimetric is a function that satisfies identity, non-negativity, and the triangle inequality, but does not require symmetry (see Section 5). By defining distances through such a function, these structural properties are enforced by construction. Consequently, once we restrict the search space to functions of the form

$$d_\theta(s, s') = d_q(\phi_\theta(s), \phi_\theta(s')),$$

the identity and triangle inequality constraints in equation 1 are satisfied automatically, and the original optimization problem simplifies to

$$\max_{\theta} \sum_{(s,s')\in\mathcal{S}^2} d_q(\phi_\theta(s), \phi_\theta(s')) \tag{2}$$

$$\text{s.t.} \quad d_q(\phi_\theta(s), \phi_\theta(s')) \leq 1 \quad \forall (s, s') \in R. \tag{One-Step}$$

This is still an idealized problem, because the one-step transition relation $R$ is typically unknown in offline data.

The remaining information needed to estimate $d_{\mathrm{MAD}}$ comes from observed trajectories. Let $\tau = (s_0, s_1, \ldots, s_T)$ be a trajectory generated by some behavior policy. For any indices $0 \leq i < j \leq T$, the difference $j - i$ is an upper bound on the Minimum Action Distance between the corresponding states: $d_{\mathrm{MAD}}(s_i, s_j) \leq j - i$, since $s_j$ is reachable from $s_i$ along the trajectory in $j - i$ decision steps.

These trajectory-derived upper bounds provide the only constraints that can be observed directly from data when the transition relation $R$ is unknown. Our learning objectives can therefore be viewed as data-driven relaxations of the simplified constrained optimization problem in equation 2, where the unobserved one-step constraints over $R$ are replaced by upper-bound constraints extracted from trajectories.

Steccanella & Jonsson (2022) learn a parameterized state embedding $\phi_\theta : \mathcal{S} \to \mathbb{R}^d$ and define a distance function $d_\theta(s, s') = d(\phi_\theta(s), \phi_\theta(s'))$, where $d$ is any distance metric in Cartesian space. The parameter vector $\theta$ of the state embedding is learned by minimizing the loss function

$$\mathcal{L} = \mathbb{E}_{\tau\sim\mathcal{D},(s_i,s_j)\sim\tau}\Big[\big(d_\theta(s_i, s_j) - (j - i)\big)^2 + w_c \cdot \mathrm{relu}\big(d_\theta(s_i, s_j) - (j - i)\big)^2\Big], \tag{3}$$

where $w_c > 0$ is a regularization factor that multiplies a penalty term which substitutes the upper bound constraints

$d_\theta(s_i, s_j) \leq j - i$. If the distance metric $d$ satisfies the triangle inequality (e.g. any norm $d = ||\cdot||_p$) then the constraints $d_\theta(s, s) = 0$ and the triangle inequality automatically hold. Enforcing the constraint $d_\theta(s_i, s_j) \leq j - i$ for each state pair $(s_i, s_j)$ on trajectories, rather than only consecutive pairs, helps learn better distance estimates, at the cost of a larger number of constraints.

## 5. Asymmetric Distance Metrics

A limitation of previous work is that the chosen distance metric $d$ is symmetric, while the MAD $d_{\text{MAD}}$ may not be symmetric. In this section, we review several asymmetric distance metrics. Concretely, a quasimetric is a function $d_q : \mathbb{R}^d \times \mathbb{R}^d \to \mathbb{R}_+$ that satisfies the following three conditions:

- **Q1** (Identity): $d_q(x, x) = 0$.

- **Q2** (Non-negativity): $d_q(x, y) \geq 0$.

- **Q3** (Triangle ineq.): $d_q(x, z) \leq d_q(x, y) + d_q(y, z)$.

A quasimetric does not require symmetry, i.e., $d_q(x, y) = d_q(y, x)$ does not hold in general.

We define a simple quasimetric $d_{\text{simple}}$ using rectified linear units:

$$d_{\text{simple}}(x, y) = \alpha \max(\text{relu}(x - y)) \\ + (1 - \alpha)d^{-1}\sum_i \text{relu}(x_i - y_i). \quad (4)$$

This metric is a weighted average of the maximum and average positive difference between the vectors $x$ and $y$ along any dimension, where $\alpha \in [0, 1]$ is a weight. In Appendix B, we show that $d_{\text{simple}}$ satisfies the triangle inequality and latent positive homogeneity (Wang & Isola, 2022).

The Wide Norm quasimetric (Pitis et al., 2020), $d_{\text{WN}}$, applies a learned transformation to an asymmetric representation of the difference between two states. The Wide Norm is defined as

$$d_{\text{WN}}(x, y) = ||W(\text{relu}(x - y) :: \text{relu}(y - x))||_2,$$

where "::" denotes concatenation and $W \in \mathbb{R}^{k \times 2d}$ is a learned weight matrix. This ensures that $d_{\text{WN}}(x, y)$ is non-negative and satisfies the triangle inequality, while concatenation is asymmetric.

The Interval Quasimetric Embedding (IQE) (Wang & Isola, 2022) leverages the Lebesgue measure of interval unions to capture asymmetric distances. IQE interprets the latent embeddings as matrices $X, Y \in \mathbb{R}^{k \times m}$ (typically obtained by reshaping a flat output vector of dimension $d = k \cdot m$). Let $x_{ij}$ denote the element in row $i$ and column $j$ of matrix

$X$. For each row $i$, we construct an interval by taking the union over the intervals defined by matrices $X$ and $Y$:

$$I_i(X, Y) = \bigcup_{j=1}^m [x_{ij}, \max\{x_{ij}, y_{ij}\}].$$

The length of this interval, denoted by $L_i(X, Y)$, is computed as its Lebesgue measure. The IQE distance is obtained by aggregating these row-wise lengths. For example, one may define

$$d_{\text{IQE}}(X, Y) = \sum_{i=1}^k L_i(X, Y),$$

or, alternatively, using a maxmean reduction:

$$d_{\text{IQE-mm}}(X, Y) = \alpha \max_{1 \leq i \leq k} L_i(X, Y) \\ + (1 - \alpha)\frac{1}{k}\sum_{i=1}^k L_i(X, Y)$$

where $\alpha \in [0, 1]$ balances the influence of the maximum and the average. This construction yields a quasimetric that inherently respects the triangle inequality while accounting for directional differences between the matrices $X$ and $Y$.

Given any of the above quasimetrics $d_q$ (i.e., $d_{\text{simple}}$, $d_{\text{WN}}$ or $d_{\text{IQE}}$), we can now define an asymmetric distance function $d_\theta(s, s') = d_q(\phi_\theta(s), \phi_\theta(s'))$. In the case of $d_{\text{IQE}}$, the state embedding $\phi : \mathcal{S} \to \mathbb{R}^d$ produces an output that is reshaped into a $k \times m$ matrix structure to parameterize the intervals. The choice of quasimetric directly shapes the trade-offs in computational cost and optimization dynamics. In Appendix E, we present an ablation study examining how this choice affects our algorithms.

## 6. Learning Asymmetric MAD Estimates

Here, we propose two novel variants of the MAD learning approach. Each trains a state encoding $\phi_\theta$ mapping states to an embedding space and uses a quasimetric $d_q$ to compute distances $d_\theta(s, s') = d_q(\phi_\theta(s), \phi_\theta(s'))$ between pairs of states $(s, s')$. Both variants support any quasimetric formulation such as $d_{\text{simple}}$, $d_{\text{WN}}$ and $d_{\text{IQE}}$, and can incorporate additional features such as gradient clipping. A full derivation of these learning objectives is provided in Appendix C.

### 6.1. MadDist: Direct Distance Learning

The first algorithm, which we call *MadDist*, learns state distances using an approach similar to prior work (Steccanella & Jonsson, 2022), but differs in the use of a quasimetric distance function and a scale-invariant loss. Concretely, MadDist minimizes the following composite loss function:

$$\mathcal{L} = \mathcal{L}_\tau + w_r\mathcal{L}_r + w_c\mathcal{L}_c. \quad (5)$$

The first loss term, $\mathcal{L}_\tau$, is a scaled version of the square difference in equation 3:

$$\mathcal{L}_\tau = \mathbb{E}_{\tau \sim \mathcal{D}, (s_i, s_j) \sim \tau} \left[ \left( \frac{d_\theta(s_i, s_j)}{j - i} - 1 \right)^2 \right]. \quad (6)$$

Crucially, scaling makes the loss invariant to the magnitude of the estimation error, which typically increases as a function of $j - i$. In other words, states that are further apart on a trajectory do not necessarily dominate the loss simply because the magnitude of the estimation error is larger.

The second loss term, $\mathcal{L}_r$, which is weighted by a factor $w_r > 0$, is a contrastive loss that encourages separation between state pairs randomly sampled from all trajectories:

$$\mathcal{L}_r = \mathbb{E}_{(s,s') \sim \mathcal{S}_\mathcal{D}} \left[ \left( \text{relu} \left( 1 - \frac{d_\theta(s, s')}{d_{\max}} \right) \right)^2 \right] \quad (7)$$

where $d_{\max}$ is a hyperparameter. Finally, the loss term $\mathcal{L}_c$, which is weighted by a factor $w_c > 0$, enforces the upper bound constraints. Specifically, let $\mathcal{D}_{\leq H_c}$ denote the set of state pairs sampled from trajectories in $\mathcal{D}$ such that the index difference satisfies $1 \leq j - i \leq H_c$ (where $H_c$ is a hyperparameter), i.e.

$$\mathcal{D}_{\leq H_c} = \{(s_i, s_j) \,|\, \tau \in \mathcal{D}, \ s_i, s_j \in \tau, \ 1 \leq j - i \leq H_c\}.$$

Then, the constraint loss is defined as:

$$\mathcal{L}_c = \mathbb{E}_{(s_i, s_j) \sim \mathcal{D}_{\leq H_c}} \left[ (\text{relu}(d_\theta(s_i, s_j) - (j - i)))^2 \right] \quad (8)$$

### 6.2. TDMadDist: Temporal Difference Learning

The second algorithm, which we call *TDMadDist*, incorporates temporal difference learning principles by maintaining a separate target embedding $\phi_{\theta'}$ and learning via bootstrapped targets. Specifically, TDMadDist learns by minimizing the loss function $\mathcal{L}' = \mathcal{L}'_\tau + w_r \mathcal{L}'_r + w_c \mathcal{L}_c$, where $\mathcal{L}_c$ is the loss term from equation 8 that enforces the upper bound constraints.

The first loss term $\mathcal{L}'_\tau$ of TDMadDist is modified to include bootstrapped distances:

$$\mathcal{L}'_\tau = \mathbb{E}_{\tau \sim \mathcal{D}, (s_i, s_j) \sim \tau} \left[ \left( \frac{d_\theta(s_i, s_j)}{\min(j - i, 1 + d_{\theta'}(s_{i+1}, s_j))} - 1 \right)^2 \right] \quad (9)$$

Hence if the current distance estimate $d_{\theta'}(s_{i+1}, s_j)$ computed using the target embedding $\phi_{\theta'}$ is smaller than $j - (i + 1)$, the objective is to make $d_\theta(s_i, s_j)$ equal to $1 + d_{\theta'}(s_{i+1}, s_j)$.

We also modify the second loss term $\mathcal{L}'_r$ to include bootstrapped distances:

$$\mathcal{L}'_r = \mathbb{E}_{\tau \sim \mathcal{D}, (s_i, s_{i+1}) \sim \tau, s_r \sim \mathcal{S}_\mathcal{D}} \left[ \left( \frac{d_\theta(s_i, s_r)}{1 + d_{\theta'}(s_{i+1}, s_r)} - 1 \right)^2 \right] \quad (10)$$

Given a state $s_i$ sampled from a trajectory of $\mathcal{D}$ and a random state $s_r \in \mathcal{S}_\mathcal{D}$, the objective is to make $d_\theta(s_i, s_r)$ equal to $1 + d_{\theta'}(s_{i+1}, s_r)$.

The target network parameters $\theta'$ are updated in each time step via an exponential moving average with hyperparameter $\beta \in (0, 1)$:

$$\theta' \leftarrow (1 - \beta)\theta' + \beta\theta. \quad (11)$$

## 7. Experiments

We evaluate our proposed MAD learning algorithms on a diverse set of environments with varying characteristics, including deterministic and stochastic dynamics, discrete and continuous state spaces, and environments with noisy observations. Our analysis is directed by the following questions:

- How accurately do our learned embeddings capture the true minimum action distances?

- How does the performance of our method compare to existing quasimetric learning approaches?

- How well does the learned distance transfer to downstream planning control?

**Code Availability.** The source code will be publicly released at: https://github.com/lorenzosteccanella/MinimumActionDistance.

**Evaluation Metrics.** We evaluate the quality of our learned representations using three metrics:

- **Spearman Correlation** ($\rho$): Measures the preservation of ranking relationships between state pairs. A high Spearman correlation indicates that if state $s_i$ is farther from state $s_j$ than from state $s_k$ in the true environment, our learned metric also predicts this same ordering. Perfect preservation of distance rankings gives $\rho = 1$.

- **Pearson Correlation** ($r$): Measures the linear relationship between predicted and true distances. A high Pearson correlation indicates that our learned distances scale proportionally with true distances (i.e. when true distances increase, our predictions increase linearly as well). Perfect linear correlation gives $r = 1$.

- **Ratio Coefficient of Variation (CV)**: Measures the consistency of our distance scaling across different state pairs. A low CV indicates that our predicted distances maintain a consistent ratio to true distances throughout the state space. For example, if we consistently predict distances that are approximately 1.5 times the true distance, CV will be low. High variation in this ratio across

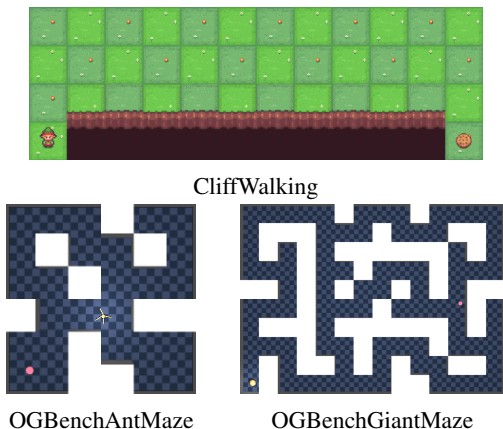

CliffWalking

OGBenchAntMaze      OGBenchGiantMaze

*Figure 2.* A subset of the environments used in our analysis.

different state pairs results in high CV. More formally, given a set of ground-truth distances $d_1, d_2, ..., d_n$ and their corresponding predicted distances $\hat{d}_1, \hat{d}_2, ..., \hat{d}_n$ where $d_i > 0$, we compute the ratios $r_i = \hat{d}_i/d_i$. The Ratio CV is given by

$$CV = \frac{\sigma_r}{\mu_r} = \frac{\sqrt{\frac{1}{n}\sum_{i=1}^{n}(r_i - \mu_r)^2}}{\frac{1}{n}\sum_{i=1}^{n} r_i}, \qquad (12)$$

**Baselines.** We compare our methods against QRL (Wang et al., 2023b), a recent quasimetric reinforcement learning approach that learns state representations using the Interval Quasimetric Embedding (IQE) formulation. QRL employs a Lagrangian optimization scheme where the objective maximizes the distance between states while maintaining locality constraints.

We also compare against PlanDist (Steccanella & Jonsson, 2022) and a variant of PlanDist that uses the simple quasimetric PlanDist-Simple.

Finally, we compare against the approach by Park et al. (2024b), an offline reinforcement learning method that embeds states into a learned Hilbert space. In this space, the distance between embedded states approximates the MAD, leading to a symmetric distance metric that cannot capture the natural asymmetry of the true MAD. We include this comparison to demonstrate the benefits of methods that explicitly model the quasimetric nature of the MAD over those that do not.

**Environments.** To evaluate the proposed methods, we designed a suite of environments where the true MAD is known, enabling a precise quantitative assessment of our learned representations. This perfect knowledge of the ground-truth distances allows us to rigorously evaluate how well different algorithms recover the underlying structure of the environment. A subset of the environments are illustrated in Figure 2, with full details provided in Appendix H.

Our test environments span a comprehensive range of MDP characteristics:

- **NoisyGridWorld**: A continuous grid world environment with stochastic transitions. The agent can move in four cardinal directions, but the action may fail with a small probability, causing the agent to remain in the same state. The initial state is random, and the goal is to reach a target state. The MAD is known and can be computed as the Manhattan distance between states. Moreover, we included random noise in the observations by extending the state $(x, y)$ with a random vector of size two, resulting in a 4-dimensional state space, where the first two dimensions are the original coordinates and the last two dimensions correspond to noise.

- **KeyDoorGridWorld**: A discrete grid world environment where the agent must find a key to unlock a door. The agent can move in four cardinal directions, and the state $(x, y, k)$ is represented by the agent's position $(x, y)$ and whether it has the key $(k)$. The MAD is known and can be computed as the Manhattan distance between states, where the distance between a state without the key and a state with the key is the sum of the distances to the key. The key can only be picked up and never dropped, creating a strong asymmetry in the distance function.

- **CliffWalking**: The original CliffWalking environment as described by Sutton & Barto (1998). The agent starts at the leftmost state and must reach the rightmost state while avoiding falling off the cliff. If the agent falls, it returns to the starting state, but the episode is not reset. This creates a strong asymmetry in the distance function, as the agent can take the shortcut by falling off the cliff to move between states.

- **PointMaze**: A continuous maze environment where the agent must navigate through a series of walls to reach a goal (Fu et al., 2020). The task in the environment is for a 2-DoF ball that is force-actuated in the Cartesian directions x and y, to reach a target goal in a closed maze. The underlying maze is a 2D grid with walls and obstacles, which we use in our experiments to approximate the ground-truth MAD by computing all-pairs shortest paths using the Floyd-Warshall algorithm on the maze graph. We consider two variants of this environment: **UMaze** and **MediumMaze**.

- **OGBench PointMaze**: A suite of physics-based maze environments that extend the standard PointMaze to much larger and more challenging layouts (Park et al., 2024c). These environments are designed to test long-horizon planning and provide two types of datasets: *navigate*, collected by a noisy expert policy navigating to ran-

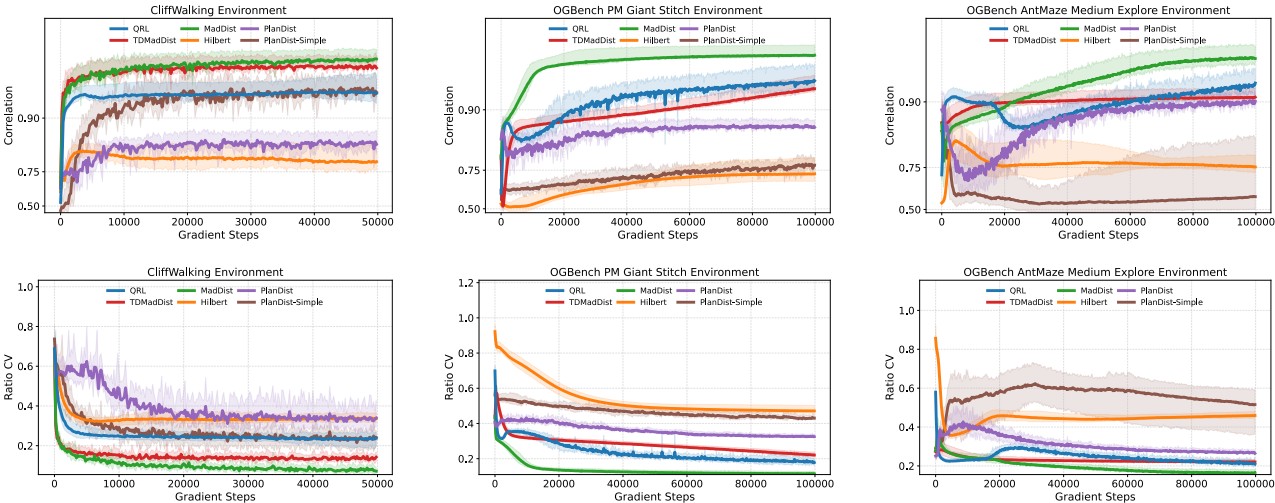

*Figure 3.* (Top) Pearson correlation coefficients and (Bottom) coefficient of variation ratios across a selection of test environments. Shaded regions show the range of values across five random seeds, with upper and lower boundaries representing maximum and minimum values.

dom goals, and *stitch*, consisting of short goal-reaching trajectories that must be combined to solve tasks.

- **OGBench AntMaze**: A high-dimensional extension of the PointMaze task where the 2-DoF ball is replaced by a complex 8-jointed quadruped ("Ant") robot controlled via torque actuators (Park et al., 2024c). It provides access to the *navigate*, *stitch* datasets, generated in the same way as in PointMaze, and an additional *explore* dataset, which consists of data collected by a purely random policy to test the agent's ability to learn from undirected transitions. The ground-truth MAD is approximated by the shortest path through the underlying maze grid.

**Empirical Setup.** We compared our two algorithms, MadDist and TDMadDist, against the QRL, Hilbert, PlanDist and PlanDist-Simple. Each method was trained for 50,000 / 100,000 / 200,000 gradient steps on offline datasets collected under different data-generation protocols. For the CliffWalking, NoisyGridWorld, and KeyDoorGridWorld environments, we used 100 trajectories collected by a random policy; for the PointMaze and AntMaze environments (including Navigate and Stitch variants), we used 1000 trajectories following the standard dataset construction of each benchmark. All reported results are means over five independent runs (random seeds) to ensure statistical robustness. For full implementation details of our evaluation setup, see Appendix D.

Figure 3 shows the Pearson correlation and coefficient of variation (CV) ratio for CliffWalking, OGBench Point-Maze Giant Maze, and the OGBench AntMaze Medium Maze environments. The full results produced in all environments, including the Spearman correlations (which we found closely matched the Pearson correlations) can

be found in Appendix G. Appendix E contains additional ablation studies, and demonstrates that MadDist and TD-MadDist are robust to the size of the latent dimension and the choice of quasimetric, and that their performance degrades gracefully with dataset size.

Table 1 reports performance on a downstream planning task, where the learned distance embeddings are used to guide the agent toward specific goals. These experiments are intended as a controlled diagnostic of representation quality rather than as a benchmark of planning performance. In much of the prior literature, learned representations are evaluated only inside full reinforcement-learning pipelines, so downstream success reflects both the quality of the representation and the details of policy learning. For example, QRL is typically paired with actor-critic optimization and additional ingredients such as behavioral cloning, making it difficult to attribute performance gains to the learned distance alone.

To mitigate these confounding factors, we deliberately use a simple random-shooting MPC planner together with the true environment simulator. This choice removes errors due to learned dynamics models and minimizes the influence of sophisticated action optimization methods, so planning performance is driven primarily by the learned distance as a measure of progress toward the goal.

This planning evaluation also complements our direct metric-learning analysis. Because our benchmark environments have known ground-truth MAD values, we can assess distance accuracy directly, which is typically impossible in standard downstream RL benchmark tasks. The planning results should therefore be interpreted as a controlled validation: if a method learns a better approximation of the MAD, that improvement should translate into better goal-reaching behavior even under a deliberately weak planner. A detailed

| Environments | PlanDist | PlanDist-Simple | QRL | TDMadDist | Hilbert | MadDist |
|---|---|---|---|---|---|---|
| AntMaze Medium Explore | $0.19 \pm 0.22$ | $0.05 \pm 0.12$ | $0.49 \pm 0.23$ | $0.39 \pm 0.25$ | $0.01 \pm 0.07$ | $\mathbf{0.80 \pm 0.21}$ |
| PointMaze Giant Navigate | $0.69 \pm 0.27$ | $0.77 \pm 0.23$ | $0.87 \pm 0.21$ | $\mathbf{0.99 \pm 0.05}$ | $0.16 \pm 0.17$ | $0.93 \pm 0.17$ |
| PointMaze Giant Stitch | $0.58 \pm 0.23$ | $0.22 \pm 0.27$ | $0.95 \pm 0.12$ | $0.74 \pm 0.26$ | $0.05 \pm 0.14$ | $\mathbf{0.99 \pm 0.07}$ |
| PointMaze Large Navigate | $0.76 \pm 0.25$ | $0.99 \pm 0.07$ | $0.97 \pm 0.09$ | $0.70 \pm 0.30$ | $0.22 \pm 0.20$ | $\mathbf{1.00 \pm 0.00}$ |
| PointMaze Large Stitch | $0.45 \pm 0.29$ | $0.17 \pm 0.23$ | $0.90 \pm 0.17$ | $0.73 \pm 0.24$ | $0.17 \pm 0.20$ | $\mathbf{1.00 \pm 0.00}$ |
| PointMaze Medium Navigate | $0.77 \pm 0.24$ | $0.89 \pm 0.17$ | $0.86 \pm 0.21$ | $0.92 \pm 0.16$ | $0.55 \pm 0.27$ | $\mathbf{1.00 \pm 0.00}$ |
| PointMaze Medium Stitch | $0.61 \pm 0.29$ | $0.51 \pm 0.29$ | $0.81 \pm 0.20$ | $0.74 \pm 0.24$ | $0.67 \pm 0.28$ | $\mathbf{1.00 \pm 0.00}$ |

*Table 1.* Success rates ($\pm$ standard deviation) across different OGBench environments. Best results per environment are shown in bold.

description of the planning setup is provided in Appendix I.

**Discussion.** From the results in Figure 3, we can see that our proposed method MadDist outperforms the QRL, Hilbert, PlanDist and PlanDist-Simple baselines in all environments, being able to learn a more accurate approximation of the MAD. Compared to QRL, this advantage likely stems from the fact that QRL primarily relies on locality constraints to shape the embedding space, while our method leverages the path distances between arbitrary states in a trajectory to form a more globally coherent representation.

The comparison with PlanDist and PlanDist-Simple is particularly informative because these methods optimize objectives that are conceptually closer to ours. PlanDist leverages trajectory-level supervision, but remains constrained by a symmetric distance formulation, limiting its ability to represent directional structure in environments with irreversible transitions. PlanDist-Simple addresses this limitation by adopting the proposed quasimetric formulation while retaining the original learning objective. The performance gap between PlanDist-Simple and MadDist therefore suggests that the gains are not explained by the quasimetric alone: the combination of asymmetric distance modelling together with our revised objective formulation, including the contrastive loss term and trajectory-based constraints (see Appendix F), is necessary to accurately recover the MAD.

Both MadDist and TDMadDist significantly outperform the Hilbert baseline, particularly in highly asymmetric environments like CliffWalking. While TDMadDist underperforms MadDist and the QRL algorithm, its performance relative to Hilbert is revealing. Since both methods utilize temporal-difference learning to propagate distance information, the superiority of TDMadDist suggests that our specific distance parametrization and learning objective provide a more effective and stable signal for approximating the MAD than the Hilbert formulation. This advantage persists even in symmetric environments, demonstrating that our approach more accurately captures the underlying geometry of the state space.

Crucially, the high accuracy of the learned distance metric directly translates to superior performance in the downstream task of goal-oriented planning, as detailed in Table 1. MadDist achieves high or perfect success rates across all environments, decisively outperforming all baselines. Its performance is particularly noteworthy in the *Stitch* environments, which require the model to compose information from disconnected trajectories, and in *AntMaze Medium Explore*, where it successfully learns from purely undirected, random data. Furthermore, its success in the large-scale *Giant* environments highlights a robust capability for long-horizon reasoning. This demonstrates that MadDist not only produces a quantitatively accurate distance function but also an effective and practical representation for planning.

## 8. Conclusion

In this paper, we present two novel algorithms for learning the Minimum Action Distance (MAD) from state trajectories. We also propose a novel quasimetric for learning asymmetric distance estimates, and introduce a set of benchmark domains with several features that make distance learning difficult. In a controlled set of experiments, we illustrate that the novel algorithms and proposed quasimetric outperform state-of-the-art algorithms for learning the MAD.

While this work has concentrated on accurately approximating the MAD as a fundamental stepping stone, it opens several promising avenues for future research. One of them is the use of MAD estimates in transfer learning and non-stationary environments, where transition dynamics evolve over time yet maintain a consistent support. On the same line, MAD can be integrated as a heuristic in search algorithms, particularly in stochastic domains, to identify the properties that make it a robust and informative guidance signal under uncertainty. Having established a reliable MAD approximation, it can now be incorporated into downstream tasks, including goal-conditioned planning and reinforcement learning, to quantify the empirical benefits it brings to complex decision-making problems.

Finally, while MAD can serve as a useful heuristic even in stochastic environments, future work will explore whether it is possible to recover the Stochastic Shortest Path Distance (SPD) or identify alternative quasimetrics that more closely align with it.

## Impact Statement

This paper presents work whose goal is to improve estimates of the minimum action distance in reinforcement learning. There are many potential societal consequences of our work, none of which we feel must be specifically highlighted here.

## Acknowledgements

Anders Jonsson is partially supported by Spanish grants PID2023-147145NB-I00 and CEX2021-001195-M, funded by MCIN/AEI/10.13039/501100011033. This work was supported by the UK Engineering and Physical Sciences Research Council (EPSRC) [EP/X025470/1].

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

# A. Proof of Uniqueness for the MAD Optimization Problem

We begin by formally defining the Minimum Action Distance (MAD) in terms of policies and minimum hitting times within a Markov Decision Process (MDP).

**Definition A.1** (Minimum Action Distance). For a fixed policy $\pi$, define the minimum hitting time from state $s$ to state $s'$ as

$$T^{\pi}_{s \to s'} = \min \left\{ t \in \mathbb{N} \cup \{\infty\} : \mathbb{P}(S_t = s' \mid S_0 = s, \pi) > 0 \right\}.$$

That is, $T^{\pi}_{s \to s'}$ denotes the earliest decision step at which state $s'$ is reachable from $s$ with non-zero probability under policy $\pi$.

The Minimum Action Distance is defined as

$$d_{\text{MAD}}(s, s') = \inf_{\pi} T^{\pi}_{s \to s'}.$$

This definition captures the earliest decision step at which a state can be reached with non-zero probability. For a fixed policy $\pi$, the quantity $T^{\pi}_{s \to s'}$ identifies the earliest reachable decision step from $s$ to $s'$. The outer infimum then selects the smallest value across all policies. In particular, if $s = s'$, then $d_{\text{MAD}}(s, s') = 0$.

**Equivalence to Graph Shortest Path**    Let $G = (\mathcal{S}, R)$ be the state-transition graph where an edge $(s, s') \in R$ exists if and only if there is an action $a$ with $P(s'|s, a) > 0$. A path of length $k$ from $s_i$ to $s_j$ in $G$ corresponds to a sequence of actions that can transition between these states with non-zero probability. We can always construct a policy $\pi$ that executes this specific sequence. Therefore, minimizing over all policies is equivalent to finding the length of the shortest path between nodes $s_i$ and $s_j$ in the graph $G$. This equivalence allows us to leverage the properties of shortest path distances in the proof below.

**Theorem A.2.** *The Minimum Action Distance, $d_{MAD}$, as defined above, is the unique solution to the constrained optimization problem:*

$$\begin{aligned}
\underset{d}{maximize} \quad & \sum_{(s,s') \in \mathcal{S}^2} d(s, s') \\
subject\ to \quad & d(s, s) = 0 \quad \forall s \in \mathcal{S} & \text{(C1)} \\
& d(s, s') \leq 1 \quad \forall (s, s') \in R & \text{(C2)} \\
& d(s, s') \leq d(s, s'') + d(s'', s') \quad \forall (s, s', s'') \in \mathcal{S}^3 & \text{(C3)}
\end{aligned}$$

*Proof.* The proof is structured in two parts. First, we show that $d_{\text{MAD}}$ is a feasible solution. Second, we show that for any other feasible solution $d$, we must have $d(s, s') \leq d_{\text{MAD}}(s, s')$, establishing both optimality and uniqueness.

**Part 1: Feasibility of $d_{\textbf{MAD}}$**

Using the shortest path interpretation of $d_{\text{MAD}}$, we verify that it satisfies each constraint.

- **Constraint (C1) - Identity:** The shortest path from any state $s$ to itself is the empty path of length 0. Thus, $d_{\text{MAD}}(s, s) = 0$.

- **Constraint (C2) - One-Step Reachability:** If $(s, s') \in R$, there exists a direct edge from $s$ to $s'$ in $G$. This corresponds to a path of length 1. The shortest path, $d_{\text{MAD}}(s, s')$, cannot be longer than this path, so $d_{\text{MAD}}(s, s') \leq 1$.

- **Constraint (C3) - Triangle Inequality:** This is a fundamental property of shortest paths. The shortest path from $s$ to $s'$ is, by definition, no longer than the path formed by concatenating the shortest path from $s$ to an intermediate state $s''$ and the shortest path from $s''$ to $s'$. This directly gives the inequality $d_{\text{MAD}}(s, s') \leq d_{\text{MAD}}(s, s'') + d_{\text{MAD}}(s'', s')$.

As $d_{\text{MAD}}$ satisfies all constraints, it is a feasible solution.

**Part 2: Optimality and Uniqueness of $d_{\textbf{MAD}}$**

Let $d$ be an arbitrary feasible solution satisfying (C1), (C2), and (C3). We show by induction on the shortest path length $k = d_{\text{MAD}}(s, s')$ that $d(s, s') \leq d_{\text{MAD}}(s, s')$.

- **Base Case ($k = 0$):** If $d_{\text{MAD}}(s, s') = 0$, then $s = s'$. By constraint (C1), any feasible solution $d$ must satisfy $d(s, s) = 0$. Thus, $d(s, s') = 0 = d_{\text{MAD}}(s, s')$.

- **Inductive Hypothesis:** Assume for some integer $k \geq 0$ that for all pairs $(s, s')$ with $d_{\text{MAD}}(s, s') \leq k$, the inequality $d(s, s') \leq d_{\text{MAD}}(s, s')$ holds.

- **Inductive Step:** Consider a pair $(s, s')$ with $d_{\text{MAD}}(s, s') = k + 1$. By the shortest path definition, there must exist a predecessor state $s''$ on a shortest path from $s$ to $s'$ such that $(s'', s') \in R$ and $d_{\text{MAD}}(s, s'') = k$.

  Applying the constraints on $d$:

$$
\begin{aligned}
d(s, s') &\leq d(s, s'') + d(s'', s') && \text{by (C3), the triangle inequality} \\
&\leq d_{\text{MAD}}(s, s'') + d(s'', s') && \text{by Inductive Hypothesis, since } d_{\text{MAD}}(s, s'') = k \\
&\leq k + 1 && \text{by (C2), since } (s'', s') \in R
\end{aligned}
$$

  Since $d_{\text{MAD}}(s, s') = k + 1$, we have shown that $d(s, s') \leq d_{\text{MAD}}(s, s')$.

By induction, we have established that for any feasible solution $d$, the inequality $d(s, s') \leq d_{\text{MAD}}(s, s')$ holds for all pairs $(s, s') \in \mathcal{S}^2$.

- **Optimality:** The objective is to maximize the sum $\sum_{(s,s') \in \mathcal{S}^2} d(s, s')$. Since we have shown that every term $d(s, s')$ is less than or equal to the corresponding term $d_{\text{MAD}}(s, s')$, the total sum for any feasible solution $d$ cannot exceed the sum for $d_{\text{MAD}}$:

$$
\sum_{(s,s') \in \mathcal{S}^2} d(s, s') \leq \sum_{(s,s') \in \mathcal{S}^2} d_{\text{MAD}}(s, s')
$$

  Since $d_{\text{MAD}}$ is itself a feasible solution, it achieves the maximum possible value, proving it is an optimal solution.

- **Uniqueness:** Let's assume $d^*$ is another solution that is also optimal.

  – For $d^*$ to be optimal, its total sum must equal the maximum possible sum:

$$
\sum_{(s,s') \in \mathcal{S}^2} d^*(s, s') = \sum_{(s,s') \in \mathcal{S}^2} d_{\text{MAD}}(s, s')
$$

  – From the induction proof we know that $d^*(s, s') \leq d_{\text{MAD}}(s, s')$ for every single pair $(s, s')$.

  Therefore $d^*(s, s') = d_{\text{MAD}}(s, s') \quad \forall (s, s') \in \mathcal{S}^2$.

$\square$

# B. Quasimetric Constructions via ReLU Reduction

Let $x, y \in \mathbb{R}^d$. We begin by defining a ReLU-based coordinate reduction, then derive scalar quasimetrics through several aggregation operators, and finally state general results for convex combinations.

## B.1. Coordinatewise ReLU Reduction

**Definition B.1** (ReLU Reduction). Define the map $r : \mathbb{R}^d \times \mathbb{R}^d \to \mathbb{R}^d$ by

$$
r(x, y) = \text{relu}(x - y), \qquad r_i(x, y) = \max\{x_i - y_i, 0\}, \quad i = 1, \ldots, d.
$$

**Proposition B.2.** *For all $x, y, z \in \mathbb{R}^d$ and $\lambda > 0$, each coordinate $r_i$ satisfies:*

*(a)* Nonnegativity and identity: $r_i(x, y) \geq 0$ *and* $r_i(x, x) = 0$.

*(b)* Asymmetry: $r_i(x, y) \neq r_i(y, x)$ *unless* $x_i = y_i$.

*(c)* Triangle inequality: $r_i(x, y) \leq r_i(x, z) + r_i(z, y)$.

*(d)* Positive homogeneity: $r_i(\lambda x, \lambda y) = \lambda\, r_i(x, y)$.

*Proof.* (a) and (b) follow directly from the definition of the max operation.

(c) Observe that

$$r_i(x, y) = \max(x_i - y_i, 0) = \max\big((x_i - z_i) + (z_i - y_i), 0\big)$$
$$\leq \max(x_i - z_i, 0) + \max(z_i - y_i, 0) = r_i(x, z) + r_i(z, y).$$

(d) Linearity of scalar multiplication inside the max gives

$$r_i(\lambda x, \lambda y) = \max(\lambda x_i - \lambda y_i, 0) = \lambda \max(x_i - y_i, 0) = \lambda r_i(x, y).$$

This concludes the proof. $\qquad\square$

## B.2. Scalar Quasimetrics via Aggregation

We now obtain real-valued quasimetrics by aggregating the vector $r(x, y)$.

**Definition B.3** (Max Reduction).
$$d_{\max}(x, y) = \max_{1 \leq i \leq d} r_i(x, y).$$

**Definition B.4** (Sum and Mean Reductions).

$$d_{\mathrm{sum}}(x, y) = \sum_{i=1}^{d} r_i(x, y), \quad d_{\mathrm{mean}}(x, y) = \tfrac{1}{d} \sum_{i=1}^{d} r_i(x, y).$$

**Proposition B.5.** *Each of $d_{\max}$, $d_{\mathrm{sum}}$, and $d_{\mathrm{mean}}$ satisfies for all $x, y, z \in \mathbb{R}^d$ and $\lambda > 0$:*

*(a)* Triangle inequality: $d(x, y) \leq d(x, z) + d(z, y)$.

*(b)* Positive homogeneity: $d(\lambda x, \lambda y) = \lambda\, d(x, y)$.

*Proof.* (a) follows by combining coordinate-wise triangle bounds with either:

- $d_{\max}$ : $\max_i[a_i + b_i] \leq \max_i a_i + \max_i b_i$,

- $d_{\mathrm{sum}}$ and $d_{\mathrm{mean}}$: term-wise summation.

(b) is immediate from the linearity of scalar multiplication and properties of max/sum. $\qquad\square$

## B.3. Convex Combinations of Quasimetrics

More generally, let $d_1, \ldots, d_n$ be any quasimetrics on $\mathbb{R}^d$ each obeying the triangle inequality and positive homogeneity. For weights $\alpha_1, \ldots, \alpha_n \geq 0$ with $\sum_k \alpha_k = 1$, define

$$d_{\mathrm{conv}}(x, y) = \sum_{k=1}^{n} \alpha_k\, d_k(x, y).$$

**Proposition B.6.** $d_{\mathrm{conv}}$ *is a quasimetric satisfying:*

*(a)* Triangle inequality: $d_{\mathrm{conv}}(x, y) \leq d_{\mathrm{conv}}(x, z) + d_{\mathrm{conv}}(z, y)$.

*(b)* Positive homogeneity: $d_{\mathrm{conv}}(\lambda x, \lambda y) = \lambda\, d_{\mathrm{conv}}(x, y)$.

*Proof.* Linearity of the weighted sum together with the corresponding property for each $d_k$ yields (a)–(b). $\qquad\square$

# C. Derivation of Learning Objectives for Minimum Action Distance

This appendix details the derivation of the *MadDist* and *TDMadDist* loss functions. The derivation begins with the foundational, but computationally intractable, constrained optimization problem for the Minimum Action Distance (MAD) and systematically transforms it into a pair of scalable learning objectives.

## C.1. Constrained Optimization Problem for MAD

The Minimum Action Distance, $d_{\text{MAD}}$, is the solution to the following constrained optimization problem. This formulation seeks a distance function that maximizes the sum of all pairwise distances while remaining consistent with the environment's one-step transition dynamics.

$$
\begin{aligned}
\underset{d}{\text{maximize}} \quad & \sum_{(s,s')\in\mathcal{S}^2} d(s,s') && \text{(Objective 1)} \\
\text{subject to} \quad & d(s,s) = 0 \quad \forall s \in \mathcal{S} && \text{(Constraint 1: Identity)} \\
& d(s,s') \leq 1 \quad \forall (s,s') \in R && \text{(Constraint 2: One-Step)} \\
& d(s,s') \leq d(s,s'') + d(s'',s') \quad \forall (s,s',s'') \in \mathcal{S}^3 && \text{(Constraint 3: Triangle Inequality)}
\end{aligned}
$$

Although the formulation uses a global maximization, the solution corresponds exactly to the *minimum* number of actions required to transition between states. The constraints enforce the correct dynamical structure:

- The one-step constraint enforces that any directly connected states must lie within distance 1, effectively *pulling* them close.

- The triangle inequality propagates this local structure globally, ensuring consistency across multi-step paths.

- The maximization objective *pushes* all state pairs as far apart as the constraints allow, yielding distances that exactly match the smallest number of steps connecting them.

This formulation is computationally intractable for large or continuous state spaces, primarily due to the triangle inequality (Constraint 3), which must hold for all triplets of states.

## C.2. Simplification via Quasimetric Embeddings

To make this problem tractable, we enforce the triangle inequality *by construction* rather than as an explicit constraint. We achieve this by learning a state embedding function $\phi : \mathcal{S} \to \mathbb{R}^k$ and defining the distance between any two states $s, s'$ using a **quasimetric** function $d_q$ on their embeddings:

$$
d_\phi(s,s') := d_q(\phi(s), \phi(s'))
$$

A quasimetric function $d_q(x,y)$ satisfies the following properties by definition:

1. **Identity:** $d_q(x,x) = 0$

2. **Non-negativity:** $d_q(x,y) \geq 0$

3. **Triangle Inequality:** $d_q(x,z) \leq d_q(x,y) + d_q(y,z)$

By defining $d_\phi$ as a quasimetric over the embedding space, the identity (Constraint 1) and triangle inequality (Constraint 3) properties are satisfied for any choice of embedding function $\phi$. This simplification is crucial, as it removes the most computationally expensive constraint and leaves us with a more manageable learning problem:

$$
\begin{aligned}
\underset{\phi}{\text{maximize}} \quad & \sum_{(s,s')\in\mathcal{S}^2} d_q(\phi(s), \phi(s')) \\
\text{subject to} \quad & d_q(\phi(s), \phi(s')) \leq 1 \quad \forall (s,s') \in R && \text{(Constraint 2: One-Step)}
\end{aligned}
$$

## C.3. The *MadDist* Loss Function

We now translate this simplified problem into a loss function suitable for minimization via gradient descent. Given a dataset of state trajectories $\mathcal{D} = \{(s_0, s_1, \ldots, s_n), \ldots\}$, the path length $j - i$ for any pair of states $(s_i, s_j)$ on a trajectory with $i < j$ provides a valid upper bound on the true MAD, i.e., $d_{\mathrm{MAD}}(s_i, s_j) \leq j - i$.

The *MadDist* loss, $\mathcal{L} = \mathcal{L}_\tau + w_r \mathcal{L}_r + w_c \mathcal{L}_c$, is composed of three terms, each corresponding to a component of the optimization problem.

**Term 1: The Trajectory Distance Loss ($\mathcal{L}_\tau$).** The original goal is to maximize all pairwise distances. As a practical proxy, we formulate a loss term that is minimized when the learned distance $d_\phi(s_i, s_j)$ matches its trajectory-based upper bound, $j - i$. This encourages the learned distances to increase, directly addressing the maximization objective by using information from the dataset. We use a scale-invariant squared error to prevent long-horizon pairs from dominating the loss.

$$\mathcal{L}_\tau = \mathbb{E}_{(s_i, s_j) \sim \mathcal{D}} \left[ \left( \frac{d_\phi(s_i, s_j)}{j - i} - 1 \right)^2 \right]$$

Minimizing $\mathcal{L}_\tau$ encourages $d_\phi(s_i, s_j) \to j - i$, serving as a proxy for the **maximize** objective.

**Term 2: The Contrastive Loss ($\mathcal{L}_r$).** To further support the global maximization objective, we introduce a contrastive term. We sample random pairs of states $(s, s')$ from the dataset and penalize them for having a small distance. This encourages all states to be far apart, which aligns with the goal of maximizing the sum of all distances, especially for pairs not on the same trajectory.

$$\mathcal{L}_r = \mathbb{E}_{(s, s') \sim \mathcal{S}_\mathcal{D}} \left[ \mathrm{relu} \left( 1 - \frac{d_\phi(s, s')}{d_{\max}} \right)^2 \right]$$

where $\mathcal{S}_\mathcal{D}$ is the set of all states appearing in the dataset $\mathcal{D}$. Minimizing $\mathcal{L}_r$ incentivizes $d_\phi(s, s')$ for random pairs to approach a large value $d_{\max}$, again serving the **maximize** objective.

**Term 3: The Constraint Loss ($\mathcal{L}_c$).** While $\mathcal{L}_\tau$ encourages matching the upper bound, it does not strictly enforce the inequality. We add an explicit penalty term that penalizes violations of the trajectory upper bound.

$$\mathcal{L}_c = \mathbb{E}_{(s_i, s_j) \sim \mathcal{D}_{<H_c}} \left[ \mathrm{relu}(d_\phi(s_i, s_j) - (j - i))^2 \right]$$

This term enforces the constraint $d_\phi(s_i, s_j) \leq j - i$, which is a generalization of the one-step constraint ($d_\phi(s, s') \leq 1$). The learning process finds an equilibrium where the objective terms ($\mathcal{L}_\tau, \mathcal{L}_r$) encourage larger distances, while this constraint term ($\mathcal{L}_c$) and the implicit triangle inequality provide regularization.

## C.4. Temporal Difference Bootstrapping (*TDMadDist*)

*TDMadDist* integrates principles from Temporal Difference (TD) learning. Instead of relying solely on the data-driven target $j - i$, it uses the model's own predictions to form a potentially tighter, more informed target. From the Bellman equation for shortest paths, we have $d_{\mathrm{MAD}}(s_i, s_j) = 1 + d_{\mathrm{MAD}}(s_{i+1}, s_j)$. We can therefore use the bootstrapped value $1 + d_{\phi'}(s_{i+1}, s_j)$ using a stable target network $\phi'$ as the new target for our objective.

The objective terms are modified as follows:

**The TD Trajectory Distance Loss ($\mathcal{L}'_\tau$).** The target for $d_\phi(s_i, s_j)$ becomes the minimum of the trajectory upper bound and the bootstrapped target.

$$\mathcal{L}'_\tau = \mathbb{E}_{(s_i, s_j) \sim \mathcal{D}} \left[ \left( \frac{d_\phi(s_i, s_j)}{\min(j - i, 1 + d_{\phi'}(s_{i+1}, s_j))} - 1 \right)^2 \right]$$

Minimizing this loss still serves the **maximize** objective, but now encourages distances toward a dynamically updated target.

**The TD Contrastive Objective ($\mathcal{L}'_r$).** The contrastive term is modified to be consistent with the one-step Bellman logic, using a bootstrapped target.

$$\mathcal{L}'_r = \mathbb{E}_{(s_i, s_r) \sim \mathcal{D}} \left[ \left( \frac{d_\phi(s_i, s_r)}{1 + d_{\phi'}(s_{i+1}, s_r)} - 1 \right)^2 \right]$$

The constraint loss $\mathcal{L}_c$ remains unchanged.

## D. Implementation Details

In this section, we describe the implementation details of each algorithm included in our evaluation. The source code will be publicly released at: https://github.com/lorenzosteccanella/MinimumActionDistance.

### D.1. Computer Resources

We run all experiments on a single NVIDIA RTX 4070 GPU with 8GB of VRAM and an Intel i7-4700-HX with 32GB of RAM.

### D.2. MadDist

To train the MadDist distance models, we used the Adam optimizer with a learning rate of $1 \times 10^{-4}$, a batch size of $256$ for the objective ($\mathcal{L}_\tau$, $\mathcal{L}_r$), and a separate batch of size $1024$ for the constraint loss ($\mathcal{L}_c$). For our main experiment, we used the novel simple quasimetric function and a latent dimension size of $512$. We include an ablation over different quasimetric functions and latent dimension sizes in Appendix E.

The full set of hyperparameter values used to train the MadDist models can be found in Table 2.

*Table 2.* Hyperparameters used to train the MadDist algorithm.

| Hyperparameter | Value |
|---|---|
| Quasimetric Function | $d_{simple}$ |
| Optimizer | AdamW (Loshchilov & Hutter, 2019) |
| Learning Rate | $1 \times 10^{-4}$ |
| Batch Size ($\mathcal{L}_\tau$, $\mathcal{L}_r$) | 256 |
| Batch Size ($\mathcal{L}_c$) | 1024 |
| Activation Function (Hidden Layers) | SELU (Klambauer et al., 2017) |
| Neural Network | (512, 512, 256) |
| $w_r$ | 10 |
| $w_c$ | 0.01 |
| $d_{max}$ | 500 |
| $H_c$ | 6 |

### D.3. TDMadDist

To train the TDMadDist distance models, we used the Adam optimizer with a learning rate of $1 \times 10^{-4}$, a batch size of $256$ for the objective ($\mathcal{L}_\tau$, $\mathcal{L}_r$), and a separate batch of size $1024$ for the constraint loss ($\mathcal{L}_c$). For our main experiment, we used the novel simple quasimetric function and a latent dimension size of $512$. We include an ablation over different quasimetric functions and latent dimension sizes in Appendix E.

For TDMadDist, we remove the hyperparameter $d_{\max}$ from the MadDist algorithm, because it is not included in TDMadDist's objective ($\mathcal{L}_r$). The temporal-difference update used when training the TDMadDist distance models involves the use of a target network, $d_{\theta'}$, which is updated using a Polyak averaging factor $\tau = 0.005$.

The full set of hyperparameter values used to train the TDMadDist models can be found in Table 3.

### D.4. QRL

We trained QRL distance models following the approach of Wang et al. (2023b). We used the Lagrangian formulation

*Table 3.* Hyperparameters used to train the TDMadDist algorithm.

| Hyperparameter | Value |
|---|---|
| Quasimetric Function | $d_{simple}$ |
| Optimizer | AdamW (Loshchilov & Hutter, 2017) |
| Learning Rate | $1 \times 10^{-4}$ |
| Batch Size ($\mathcal{L}_\tau, \mathcal{L}_r$) | 256 |
| Batch Size ($\mathcal{L}_c$) | 1024 |
| Activation Function (Hidden Layers) | SELU (Klambauer et al., 2017) |
| Neural Network | (512, 512, 256) |
| $w_r$ | 1 |
| $w_c$ | 0.01 |
| $H_c$ | 6 |
| $\tau$ | 0.005 |

$$\min_{\theta} \max_{\lambda \geq 0} -\mathbb{E}_{s,s' \sim S_D}[\phi(d_\theta^{\text{IQE}}(s, s'))] + \lambda \left( \mathbb{E}_{(s,s') \sim p_{\text{transition}}}[\text{relu}(d_\theta^{\text{IQE}}(s, s') + 1)^2] \right), \tag{13}$$

where $\phi(x) \triangleq -\text{softplus}(\alpha - x, \beta)$ and $d_\theta^{\text{IQE}}(s, s')$ is the IQE distance between states $s$ and $s'$. Following Wang et al. (2023b), we set $(\alpha, \beta) = (15, 0.1)$ for short-horizon environments and $(\alpha, \beta) = (500, 0.01)$ for long-horizon environments. The first term in the objective maximizes the expected distance between states sampled from the dataset, while the second term penalizes distances between state–next-state pairs $(s, s')$ observed in the data.

For the neural network architecture, we used a multi-layer perceptron with an overall layer structure of $x$ - 512 - 512 - 128 (where $x$ is the input observation dimension). Its two hidden layers (each of size 512) use ReLU activations, as described for state-based observations environments (i.e., environments with real vector observations, as opposed to images or other high-dimensional inputs) in the original paper. For the distance function, the resulting 128-dimensional MLP output is fed into a separate 128-512-2048 projector, followed by an IQE-maxmean head with 64 components each of size 32.

The full set of hyperparameter values used to train the QRL distance models can be found in Table 4.

*Table 4.* Hyperparameters used to train the QRL model.

| Hyperparameter | Value |
|---|---|
| Neural Network State embedding | $x$ - 512 - 512 - 128 |
| Neural Network IQE Projector | 128-512-2048 |
| Activation Function (Hidden Layers) | ReLU (Glorot et al., 2011) |
| Optimizer | Adam (Kingma & Ba, 2015) |
| $\lambda$ Learning Rate | 0.01 |
| Learning Rate Model | $1 \times 10^{-4}$ |
| Batch Size | 1024 |
| Quasimetric function | IQE |
| IQE n components | 64 |
| IQE Reduction | maxmean |

## D.5. Hilbert Representation

A Hilbert representation model is a function $\phi : \mathcal{S} \to \mathbb{R}^d$ that embeds a state $s \in \mathcal{S}$ into a $d$-dimensional space, such that the Euclidean distance between embedded states approximates the number of actions required to transition between them under the optimal policy.

We trained Hilbert representation models following the approach of Park et al. (2024b), using action-free Implicit Q-Learning (IQL) (Park et al., 2023) and Hindsight Experience Replay (HER) (Andrychowicz et al., 2017).

We used a dataset of state–next-state pairs $(s, s')$, which we relabeled using HER to produce state–next-state–goal tuples $(s, s', g)$. Goals were sampled from a geometric distribution $\text{Geom}(\gamma)$ over future states in the same trajectory with

probability $0.625$, and uniformly from the entire dataset with probability $0.375$.

We trained the Hilbert representation model $\phi$ to minimize the temporal-difference loss

$$\mathbb{E}[l_\tau(-\mathbf{1}(s \neq g) - \gamma||\phi(s') - \phi(g)|| + ||\phi(s) - \phi(g)||)], \tag{14}$$

where $l_\tau$ denotes the expectile loss (Newey & Powell, 1987), an asymmetric loss function that approximates the $\max$ operator in the Bellman backup (Kostrikov et al., 2022). This objective naturally supports the use of target networks (Mnih et al., 2015) and double estimators (Van Hasselt et al., 2016) to improve learning stability. We included both in our implementation, following the original setup used by Park et al. (2024b).

The full set of hyperparameter values used to train the Hilbert models can be found in Table 5.

*Table 5.* Hyperparameters used to train the Hilbert representation models.

| Hyperparameter | Value |
|---|---|
| Latent Dimension | 32 |
| Expectile | 0.9 |
| Discount Factor | 0.99 |
| Learning Rate | 0.0003 |
| Target Network Smoothing Factor | 0.005 |
| Multi-Layer Perceptron Dimensions | (512, 512) Fully-Connected Layers |
| Activation Function (Hidden Layers) | GELU (Hendrycks & Gimpel, 2016) |
| Layer Normalization (Hidden Layers) | True |
| Activation Function (Final Layer) | Identity |
| Layer Normalization (Final Layer) | False |
| Optimizer | Adam (Kingma & Ba, 2015) |
| Batch Size | 1024 |

## D.6. PlanDist

We trained PlanDist models following the approach of Steccanella & Jonsson (2022). As described in Section 4, PlanDist learns a state embedding $\phi_\theta$ by minimizing the discrepancy between learned distances and trajectory distances $(j - i)$, while enforcing upper-bound constraints through a penalty term.

For the neural network architecture, we used a multi-layer perceptron with hidden layer sizes (512, 512, 256) and SELU activations (Klambauer et al., 2017), matching the architecture used for MadDist and TDMadDist. We trained the models using the AdamW optimizer with a learning rate of $1 \times 10^{-4}$ and a batch size of 256. We set the constraint weight factor to $w_c = 0.01$.

For the standard PlanDist baseline, we used the L1 (Manhattan) distance in latent space. In addition, we evaluated a variant of PlanDist using the simple quasimetric function ($d_{\text{simple}}$) in place of the L1 distance.

The full set of hyperparameter values used to train the PlanDist models can be found in Table 6.

*Table 6.* Hyperparameters used to train the PlanDist models.

| Hyperparameter | Value |
|---|---|
| Distance Function | L1 / $d_{simple}$ |
| Optimizer | AdamW (Loshchilov & Hutter, 2019) |
| Learning Rate | $1 \times 10^{-4}$ |
| Batch Size | 256 |
| Activation Function (Hidden Layers) | SELU (Klambauer et al., 2017) |
| Neural Network | (512, 512, 256) |
| $w_c$ | 0.01 |

# E. Ablation Study

In this section, we present additional ablation studies to analyze the performance of our proposed methods. We evaluate the impact of different hyperparameters and design choices on the performance of the learned embeddings.

We conduct experiments in the CliffWalking environment, which is a highly asymmetric environment with a known ground-truth MAD. For each experiment we train the *MadDist* algorithm using the same hyperparameters from the main experiments, varying only the hyperparameter of interest while keeping all others fixed. We then evaluate the learned embeddings using Spearman correlation, Pearson correlation, and Ratio CV metrics.

## E.1. Effect of Latent Dimension on MAD Accuracy

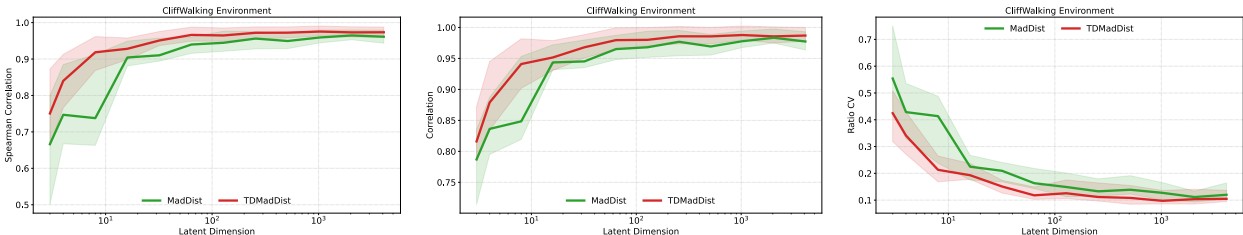

*Figure 4.* Impact of latent size on Spearman correlation, Pearson correlation and Ratio CV of the MadDist and TDMadDist algorithms, evaluated in the CliffWalking environment. Shaded regions show the range of values across five random seeds, with upper and lower boundaries representing maximum and minimum values.

Figure 4 shows the impact of the latent dimension size on the performance of our proposed methods. We can see that increasing the latent dimension size improves the performance of our methods. We note that the performance starts to saturate after a latent dimension size of 10, but larger latent dimension sizes still slightly improve the performance and do not harm the performance. This is likely due to the fact that larger latent dimension sizes allow for more expressive representations, which can help to better capture the underlying structure of the environment.

## E.2. Effect of Quasimetric Choice on MAD Accuracy

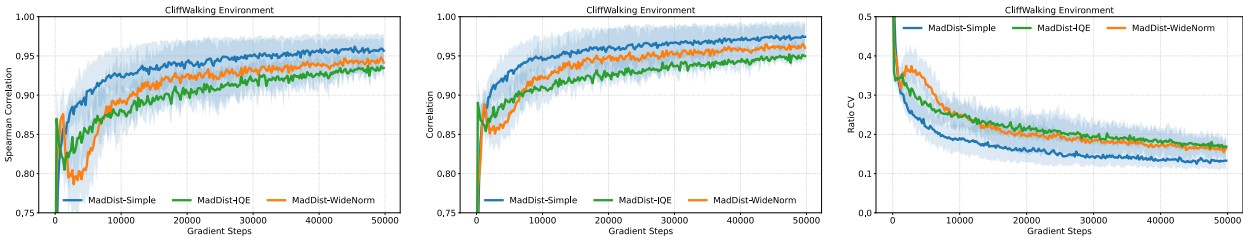

*Figure 5.* Impact of different quasimetric functions on correlation and Ratio CV of the MadDist algorithm, evaluated in the CliffWalking environment. Shaded regions show the range of values across five random seeds, with upper and lower boundaries representing maximum and minimum values.

Figure 5 shows the impact of different quasimetric functions on the performance of the learned MadDist model. The novel simple quasimetric (MadDistance-Simple) achieves the best performance, outperforming both the Wide Norm (MadDistance-WideNorm) and IQE (MadDistance-IQE) variants. While Wide Norm and IQE perform similarly to each other, they consistently underperform the simple quasimetric across all three evaluation metrics.

Figure 6 presents the same ablation over quasimetric functions, now applied to learning the TDMadDist model. The results mirror the previous setting: the simple quasimetric (TDMadDist-Simple) again achieves the strongest performance, while the Wide Norm (TDMadDist-WideNorm) and IQE (TDMadDist-IQE) variants lag slightly behind and show comparable results to each other.

In this experiment, we used a latent dimension size of 256. For the Wide Norm quasimetric, we configure the model with 32 components, each having an output component size of 32. For the IQE quasimetric, we set each component to have a

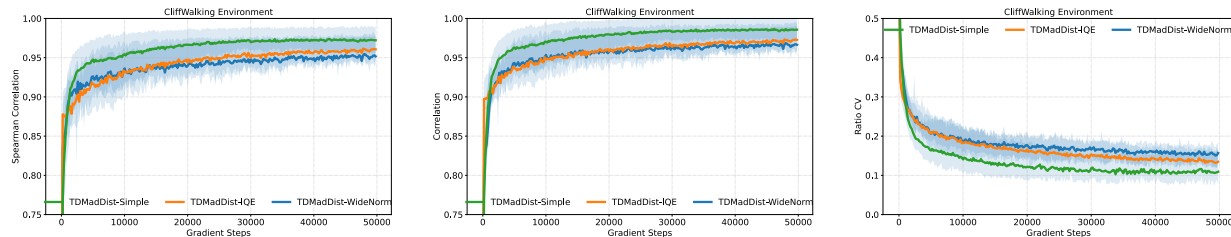

*Figure 6.* Impact of different quasimetric functions on correlation and Ratio CV of the TDMadDist algorithm, evaluated in the CliffWalking environment. Shaded regions show the range of values across five random seeds, with upper and lower boundaries representing maximum and minimum values.

dimensionality of 16. For both quasimetric functions we use maxmean reduction (Pitis et al., 2020).

### E.3. Effect of Dataset Size on MAD Accuracy

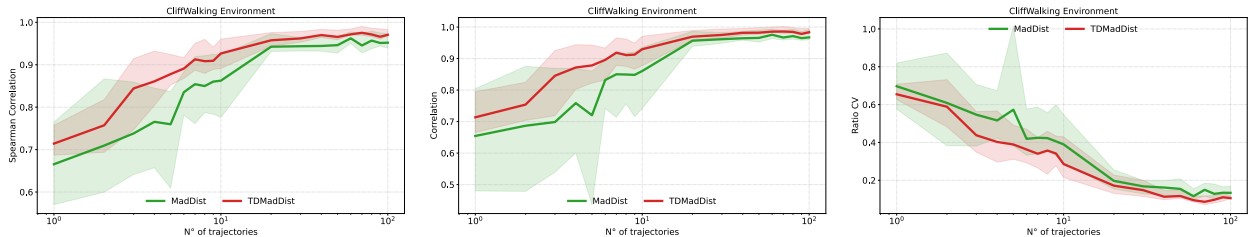

*Figure 7.* Impact of dataset size on Spearman correlation, Pearson correlation and Ratio CV of the MadDist and TDMadDist algorithms, evaluated in the CliffWalking environment. Shaded regions show the range of values across five random seeds, with upper and lower boundaries representing maximum and minimum values.

Figure 7 illustrates how dataset size affects the performance of our proposed methods. As the number of trajectories increases, the dataset provides broader coverage of all the possible transitions in the environment, leading to a more accurate approximation of the MAD.

### E.4. Neural Network Size Choice for QRL and Hilbert

In this section, we present ablation studies examining how the size of the neural network affects performance for both QRL and Hilbert. For QRL, we evaluate three architectures, each consisting of an embedding network followed by a projection network used with the IQE quasimetric as described in (Wang et al., 2023b):

- QRL_nn_1: (512, 512, 128) embedding + (128, 512, 2048) projection.

- QRL_nn_2: (512, 512, 256, 128) embedding + (128, 512, 2048) projection.

- QRL_nn_3: (1024, 1024, 256) embedding + (1024, 1024, 1024, 2048) projection.

QRL_nn_1 corresponds to the architecture used for state-based observations in (Wang et al., 2023b), while QRL_nn_2 shares the same embedding network as MAD and TDMAD. QRL_nn_3 represents the larger architecture considered in (Wang et al., 2023b). As shown in Figure 8, performance differences across these architectures are minor, with QRL_nn_1 achieving the best results.

For the Hilbert algorithm, we compare two fully connected architectures:

- HILBERT_nn_1: (512, 512), as used in the original paper (Park et al., 2024b).

- HILBERT_nn_2: (512, 512, 256, 128), matching the architecture used for MAD and TDMAD.

As shown in Figure 9, HILBERT_nn_1 performs best in our evaluation.

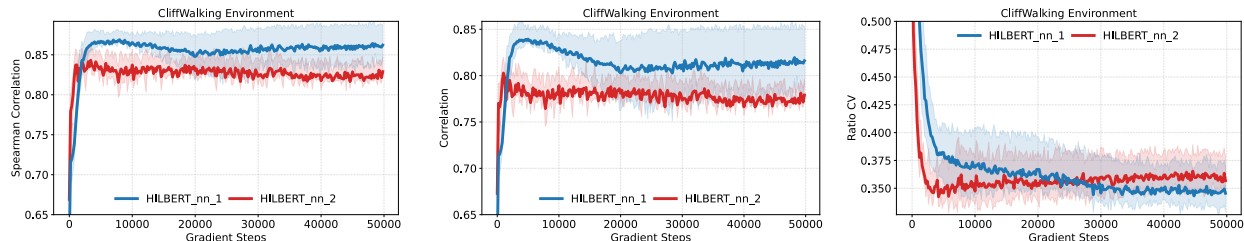

*Figure 8.* Impact of neural network size on Spearman correlation, Pearson correlation, and Ratio CV for the QRL algorithm in the CliffWalking environment. Shaded regions show the range of values across five random seeds, with upper and lower boundaries representing maximum and minimum values.

*Figure 9.* Impact of neural network size on Spearman correlation, Pearson correlation, and Ratio CV for the Hilbert algorithm in the CliffWalking environment. Shaded regions show the range of values across five random seeds, with upper and lower boundaries representing maximum and minimum values.

# F. Ablation Study on the Constraint Horizon and Contrastive Weight

We performed an ablation study analysing the effect of the constraint horizon parameter ($H_c$) and the contrastive regularization weight ($w_r$) across two OGBench environments: AntMaze Navigate, which uses expert-generated trajectories, and AntMaze Explore, which uses trajectories collected by a random policy.

Figures 10 and 11 report the relative performance change as a function of the contrastive weight ($w_r$). Across both environments, increasing ($w_r$) consistently improves performance over the baseline setting ($w_r = 0$), indicating that the contrastive objective provides a complementary learning signal beyond ($L_\tau$).

The gains are substantially larger in AntMaze Explore, where the trajectories are less structured and contain weaker supervision signals. In this setting, larger values of ($w_r$) lead to stronger and more consistent improvements across all ground-truth distance ranges. In contrast, in AntMaze Navigate, where trajectories already contain coherent expert behavior, the improvements remain positive but comparatively smaller. These results suggest that the contrastive objective is particularly valuable in low-quality or highly stochastic data regimes, where trajectory information alone is insufficient to accurately recover the geometry of the state space.

We additionally evaluate the influence of the constraint horizon parameter ($H_c$). Figure 12 shows the corresponding ablation on AntMaze Explore. Increasing ($H_c$) consistently improves performance, particularly for state pairs separated by shorter ground-truth distances. This suggests that enforcing upper-bound consistency over a larger subset of trajectory pairs helps stabilize the local geometric structure of the learned embedding and improves the propagation of distance information across the representation space.

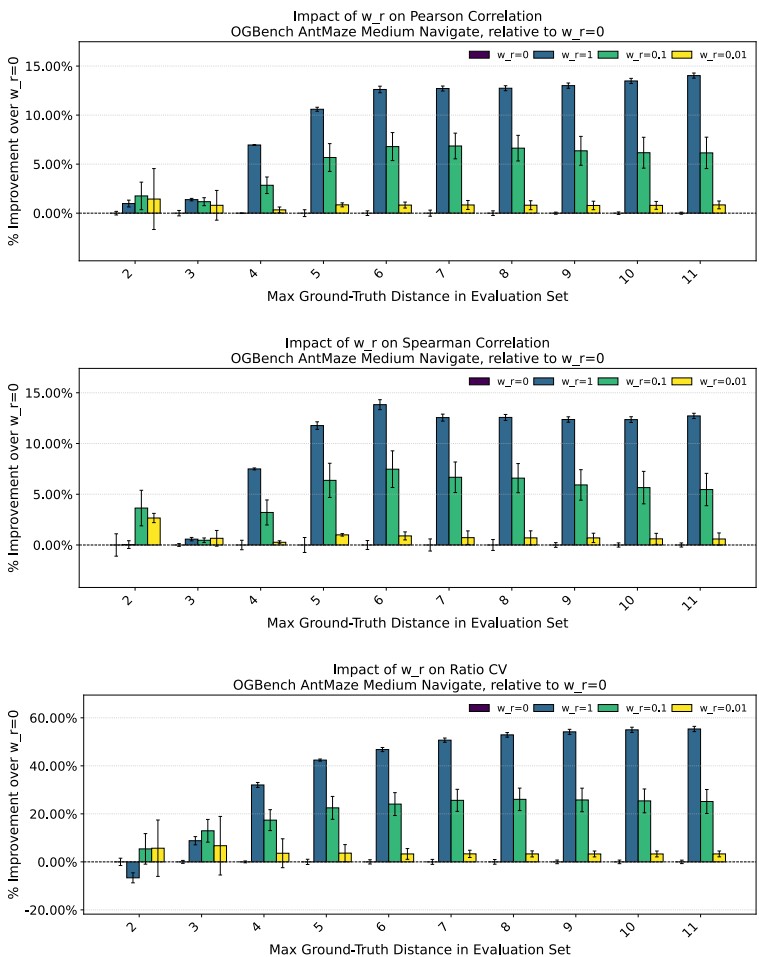

*Figure 10.* Impact of $w_r$ on Pearson and Spearman correlation coefficients and coefficient of variation (CV) ratios on ogbench AntMaze Medium Navigate

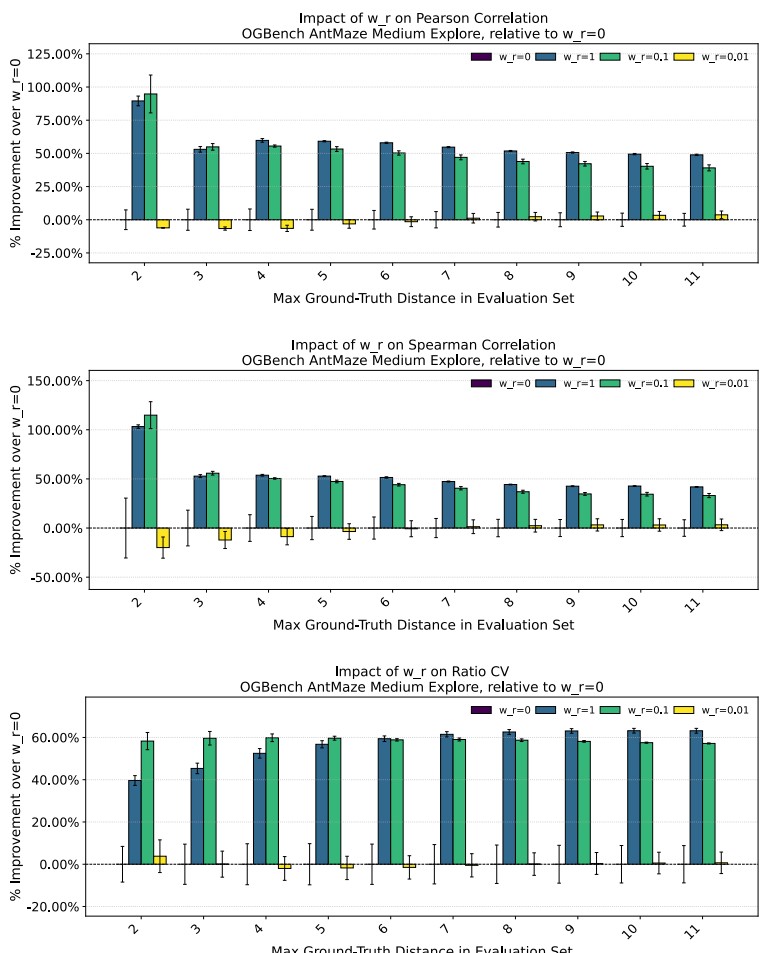

*Figure 11.* Impact of $w_r$ on Pearson and Spearman correlation coefficients and coefficient of variation (CV) ratios on ogbench AntMaze Medium Explore

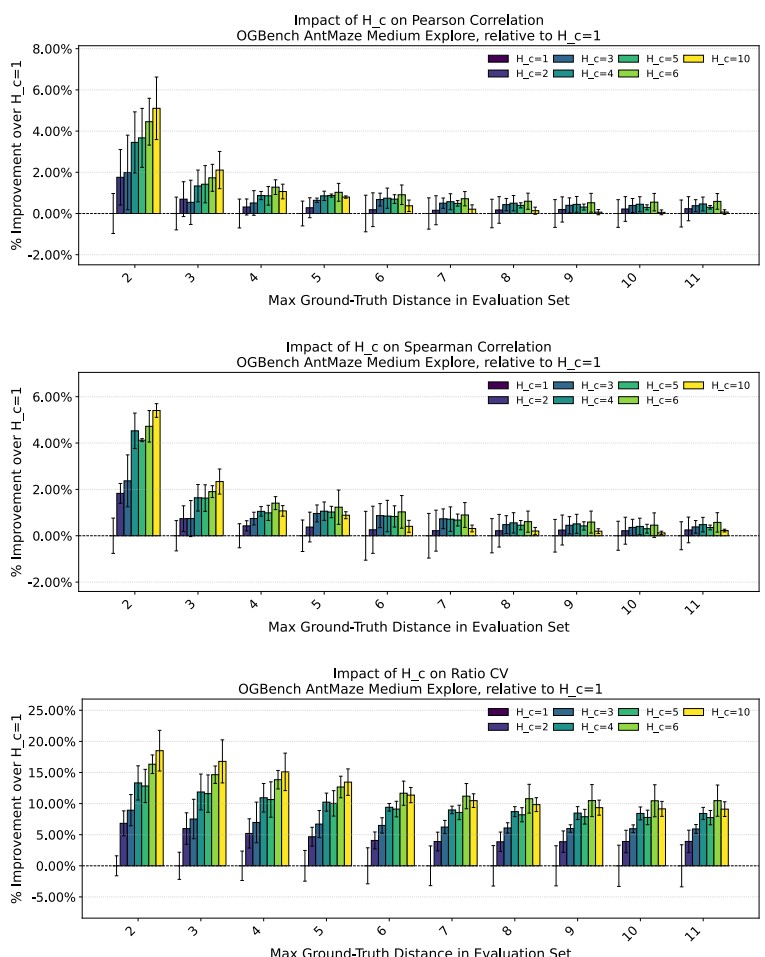

*Figure 12.* Impact of $H_c$ on Pearson and Spearman correlation coefficients and coefficient of variation (CV) ratios on ogbench AntMaze Medium Explore

# G. Complete list of results

In this section, we present the complete list of results produced across the full suite of evaluation environments included in our study. We also present a qualitative analysis of the MAD metric learned in the MediumMaze environment.

## G.1. Qualitative Evaluation

In this section, we present a qualitative evaluation of the MadDist algorithm in the MediumMaze environment by visualizing the learned geometry of the state space to demonstrate how the metric captures the underlying structure of the maze. As shown in Figure 13, the learned metric aligns well with the structure induced by the maze's walls and faithfully captures the minimum action distance between points over both short and long horizons.

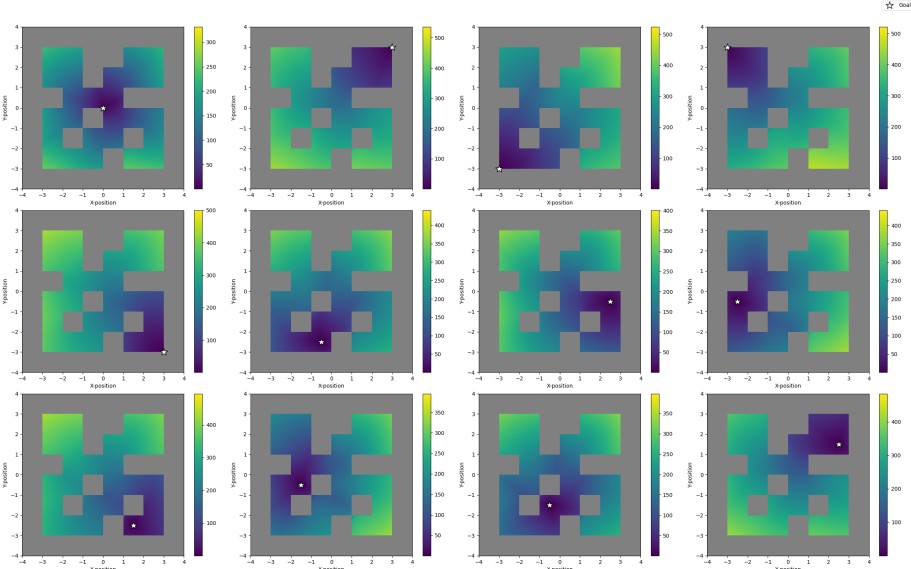

*Figure 13.* Visualization of the learned MAD landscape using MadDist on the MediumMaze environment. The heatmap represents the predicted distance from a fixed goal state to every other point in the maze

## G.2. Full Distance Learning Results

In this section, we present the full set of results evaluating how accurately the proposed MadDist and TDMadDist approaches learn the minimum action distance across all environments used in our evaluation, compared to the QRL and Hilbert baselines. Accuracy is measured using Spearman and Pearson correlation coefficients, together with the Ratio Coefficient of Variation (Ratio CV). This full set of results can be seen in Figures 14, 15, and 16.

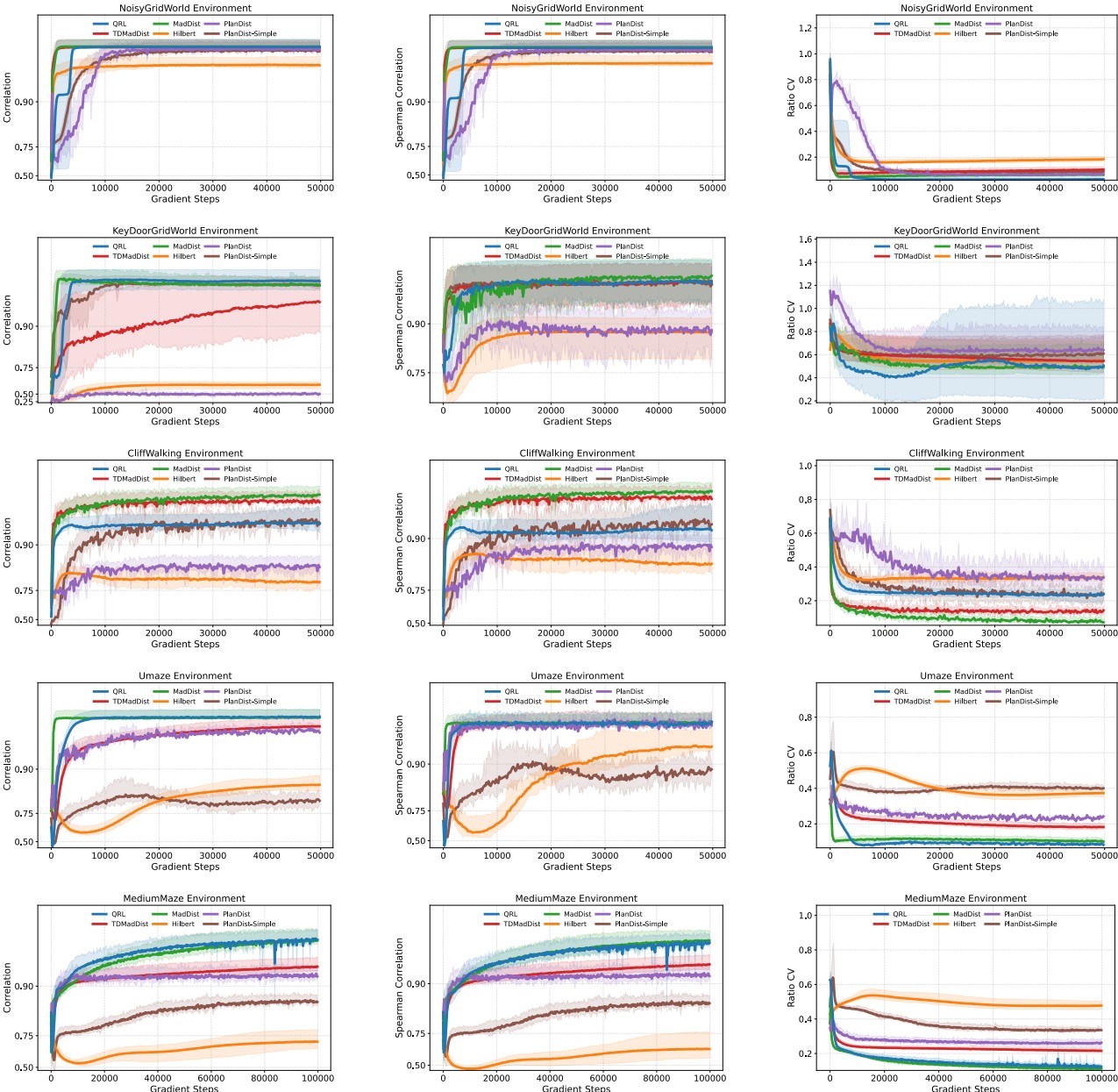

*Figure 14.* Pearson and Spearman correlation coefficients and coefficient of variation (CV) ratios across the NoisyGridWorld, KeyDoor-GridWorld, CliffWalking, UMaze, and MediumMaze environments. Shaded regions show the range of values across five random seeds, with upper and lower boundaries representing maximum and minimum values.

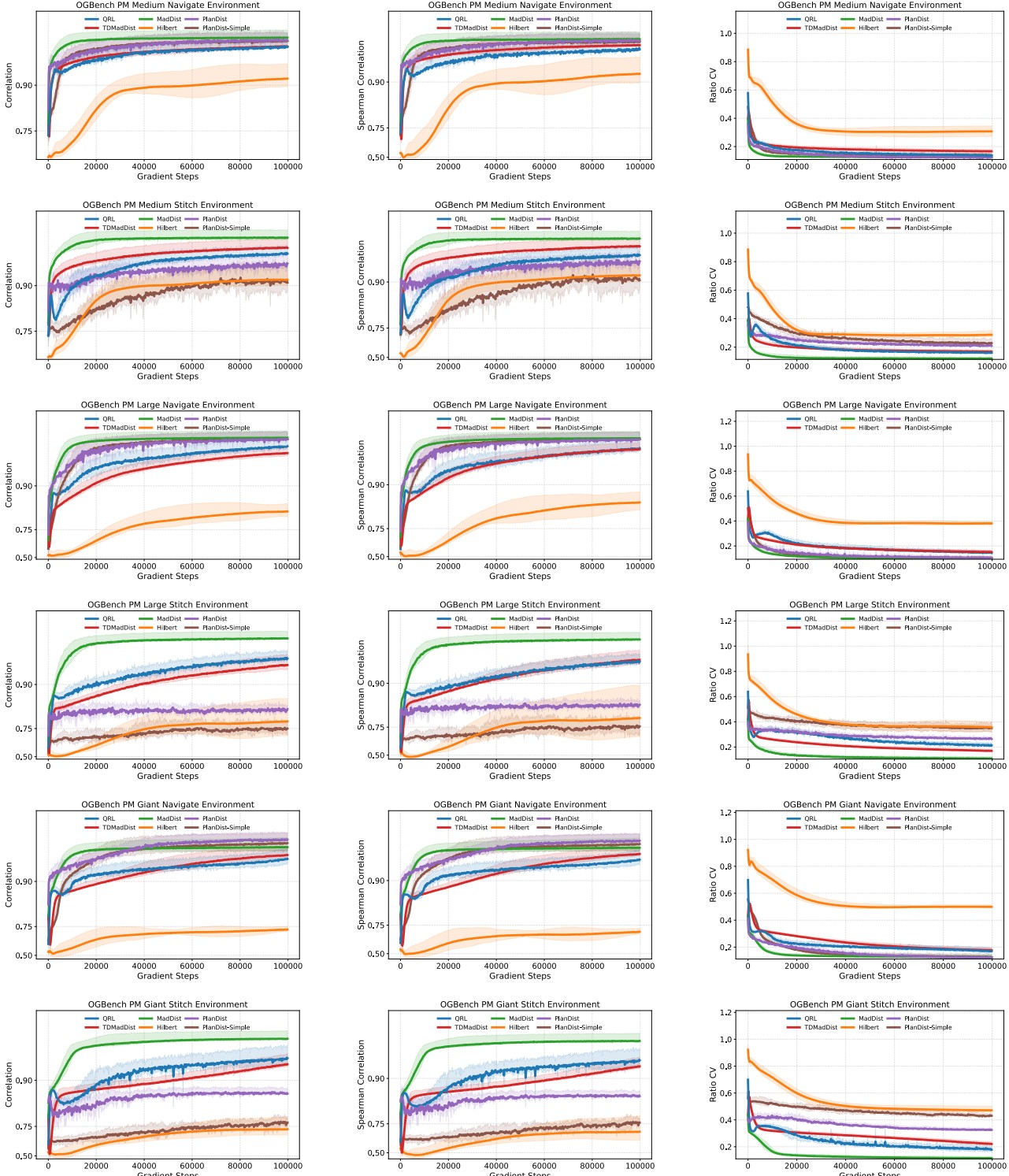

*Figure 15.* Pearson and Spearman correlation coefficients and coefficient of variation (CV) ratios across OGBench PointMaze test environments. Shaded regions show the range of values across five random seeds, with upper and lower boundaries representing maximum and minimum values.

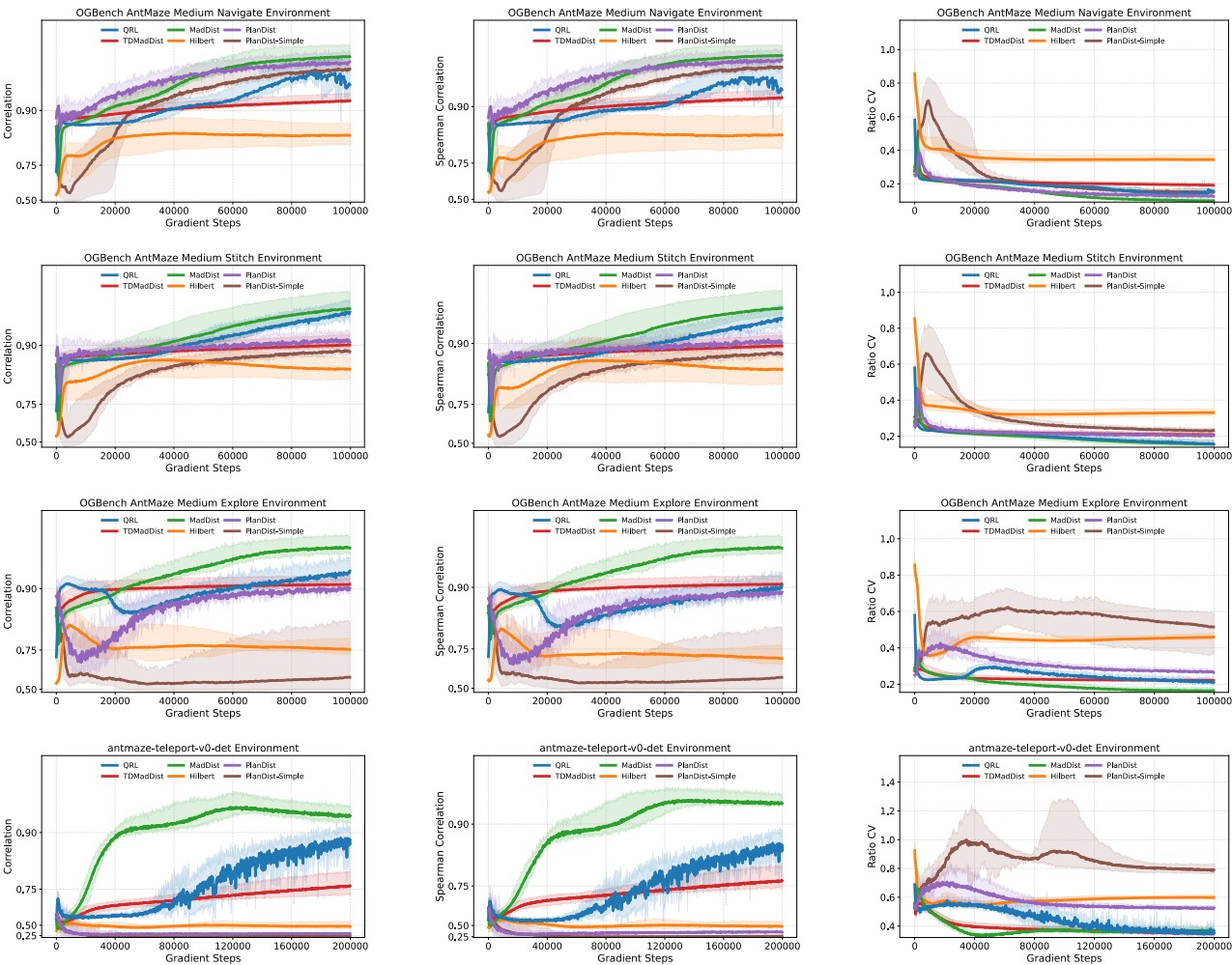

*Figure 16.* Pearson and Spearman correlation coefficients and coefficient of variation (CV) ratios across OGBench AntMaze test environments. Shaded regions show the range of values across five random seeds, with upper and lower boundaries representing maximum and minimum values.

# H. Environments

Our test environments were specifically chosen to span a comprehensive range of reward-free MDP characteristics and challenges, ensuring a thorough evaluation. Key design considerations for this suite include:

- *Noisy Observations:* To assess robustness to imperfect state information, which can challenge algorithms relying on precise state identification.

- *Stochastic Dynamics:* To evaluate if our algorithm can retrieve the MAD even when transitions are not deterministic. This reflects real-world scenarios where environments have inherent randomness or agent actions have uncertain outcomes.

- *Asymmetric Transitions:* To test the capability of our algorithm to learn true quasimetric distances that capture directional dependencies (e.g., one-way paths, key-door mechanisms).

- State Spaces:

  - *Continuous State Spaces:* To demonstrate applicability to problems with real-valued state representations where function approximation is essential.
  - *Discrete State Spaces:* To provide foundational testbeds with clearly defined structures and allow for exact MAD computation.

- Action Spaces:

  - *Continuous Action Spaces:* To evaluate performance in environments where actions are defined by real-valued parameters, common in robotics and physical control tasks.
  - *Discrete Action Spaces:* To ensure applicability to environments with a finite set of distinct actions.

- *Complex Dynamics:* Incorporating environments like PointMaze, which feature non-trivial physics (velocity, acceleration).

- *Hard Exploration:* Utilizing environments with complex structures (e.g., intricate mazes) that pose significant exploration challenges for naive data collection policies (like the random policy we used in our experiments).

### NoisyGridWorld

*Noisy Observations, Stochastic Dynamics, Continuous State Space, Discrete Action Space*

- **State space:** The agent receives a 4-dimensional observation vector $(x, y, n_1, n_2)$ at each step. In this observation, $(x, y)$ are discrete coordinates in a $13 \times 13$ grid, and $(n_1, n_2) \sim \mathcal{N}(0, \sigma^2 I)$ are i.i.d. Gaussian noise components. The true underlying latent state, which is not directly observed by the agent in its entirety without noise, is the coordinate pair $(x, y)$. The presence of the noise components $(n_1, n_2)$ in the observation makes the sequence of observations non-Markovian with respect to this true latent state.

- **Action space:** Four stochastic actions are available in all states: UP, DOWN, LEFT, and RIGHT.

- **Transition dynamics:** With probability 0.5, the intended action is executed; with probability 0.5, a random action is applied. Transitions are clipped at grid boundaries.

- **Initial state distribution ($\mu_0$):** The agent's initial true latent state $(x_0, y_0)$ is a random real-valued position sampled uniformly from the grid. The full initial observation is $(x_0, y_0, n_{1,0}, n_{2,0})$, where the initial noise components $(n_{1,0}, n_{2,0})$ are also sampled i.i.d. from $\mathcal{N}(0, \sigma^2 I)$. The real-valued nature of both the initial position and the noise components makes the observed state space continuous.

- **Ground-truth MAD:** Since the latent state is deterministic apart from noise, the MAD between two states $(x_1, y_1)$ and $(x_2, y_2)$ is the Manhattan distance $|x_1 - x_2| + |y_1 - y_2|$. Noise components are ignored.

**KeyDoorGridWorld**

*Asymmetric, Deterministic Dynamics, Discrete State Space, Discrete Action Space*

- **State space:** States are triples $(x, y, k)$, where $(x, y)$ is the agent's position in a $13 \times 13$ grid, and $k \in \{0, 1\}$ indicates whether the key has been collected.

- **Action space:** Four deterministic actions are available in all states: UP, DOWN, LEFT, and RIGHT.

- **Transition dynamics:** Transitions are deterministic. The agent picks up the key by visiting the key's cell; the key cannot be dropped once collected. The door can only be passed if the key has been collected.

- **Initial state distribution ($\mu_0$):** The agent starts at position $(1, 1)$.

- **Ground-truth MAD:** Defined as the minimum number of steps to reach the target state, accounting for key dependencies. For example, if the agent lacks the key and the goal requires it, the path must include visiting the key first.

**CliffWalking**

*Asymmetric, Deterministic Dynamics, Discrete State Space, Discrete Action Space*

- **State space:** The environment is a $4 \times 12$ grid. Each state corresponds to a discrete cell $(x, y)$.

- **Action space:** Four deterministic actions are available in all states: UP, DOWN, LEFT, or RIGHT.

- **Transition dynamics:** Transitions are deterministic unless the agent steps into a cliff cell, in which case it is returned to the start. The episode is not reset.

- **Initial state distribution ($\mu_0$):** The agent starts at position $(1, 1)$.

- **Ground-truth MAD:** The MAD is the minimal number of steps required to reach the target state, allowing for cliff transitions. Since falling into the cliff resets the agent's position, it can create shortcuts and lead to strong asymmetries in the distance metric.

**PointMaze**

*Continuous State Space, Complex Dynamics, Hard exploration, Continuous Action Space*

- **State space:** The agent observes a 4-dimensional vector $(x, y, \dot{x}, \dot{y})$, where $(x, y)$ is the position of a green ball in a 2D maze and $(\dot{x}, \dot{y})$ are its linear velocities in the $x$ and $y$ directions, respectively.

- **Action space:** Continuous control inputs $(a_x, a_y)$ corresponding to applied forces in the $x$ and $y$ directions. The applied force is limited to the range $[-1, 1]$ N in each direction.

- **Transition dynamics:** The system follows simple force-based dynamics within the MuJoCo physics engine. The applied forces affect the agent's velocity, which in turn updates its position. The ball's velocity is limited to the range $[-5, 5]$ m/s in each direction. Collisions with the maze's walls are inelastic: any attempted movement through a wall is blocked.

- **Initial state distribution ($\mu_0$):** The agent starts at a random real-valued position $(x, y)$ sampled uniformly from valid maze locations. The initial velocities $(\dot{x}_0, \dot{y}_0)$ are set to $(0, 0)$.

- **Ground-truth MAD:** The maze is discretized into a uniform grid. Using the Floyd-Warshall algorithm on the resulting connectivity graph, we compute shortest path distances between all reachable pairs of positions.

**OGBench PointMaze**

*Continuous State Space, Complex Dynamics, Hard Exploration, Continuous Action Space*

This benchmark extends the `PointMaze` environment to significantly larger and more challenging mazes, designed to test long-horizon reasoning and exploration capabilities. The controlled agent is the same 2D ball as in `PointMaze`, but the scale and complexity of the mazes increase substantially.

- **Medium:** Matches the original medium maze from D4RL.

- **Large:** Matches the original large maze from D4RL.

- **Giant:** Twice the size of `Large`, with a layout adapted from the `antmaze-ultra` maze of (Jiang et al., 2022). It contains longer paths, requiring up to 1000 environment steps, making it especially demanding for long-horizon planning.

Two datasets are provided for each maze:

- **Navigate:** Collected using a noisy expert policy that repeatedly navigates to randomly sampled goals throughout the maze.

- **Stitch:** Consists of short, goal-reaching trajectories of at most 4 cell units in length. Solving tasks requires stitching together multiple short demonstrations (up to 8), testing the agent's ability to compose behaviors across long horizons.

**OGBench AntMaze**

*Continuous State Space, Complex Dynamics, Hard Exploration, Continuous Action Space*

This environment represents a high-dimensional extension of the `PointMaze` task, replacing the 2D ball with a complex 8-jointed quadruped ("Ant") robot. It combines the long-horizon navigation challenges of large-scale mazes with the difficult locomotion dynamics of a torque-controlled legged robot.

- **State space:** The agent receives a 29-dimensional observation vector, which includes the Cartesian position and velocity of the torso, as well as the angles and velocities of the eight leg joints.

- **Action space:** An 8-dimensional continuous action space corresponding to the torque applied to each of the quadruped's joints.

- **Transition dynamics:** The environment uses the MuJoCo physics engine to simulate the complex contact dynamics and articulation of the Ant robot. Successful movement requires the agent to coordinate joint torques to maintain balance and achieve locomotion while navigating around obstacles.

- **Datasets:** Three distinct datasets are provided to evaluate different aspects of offline learning:
  - **Navigate:** Generated by a noisy expert policy navigating to random goals.
  - **Stitch:** Composed of short, localized goal-reaching trajectories that must be concatenated to solve long-horizon tasks.
  - **Explore:** Collected using a purely random policy. This dataset contains no goal-directed behavior, serving as a rigorous test for an agent's ability to extract the underlying distance metric from undirected exploration data.

- **Ground-truth MAD:** Similar to the PointMaze environments, the ground-truth MAD is approximated by discretizing the maze layout and calculating the shortest-path distances through the grid using the Floyd-Warshall algorithm.

**OGBench AntMaze Teleport**

*Continuous State Space, Complex Dynamics, Asymmetric Transitions, Continuous Action Space*

This environment extends `OGBench AntMaze` by introducing deterministic *teleportation*: at two designated trigger cells (*black holes*), the agent is instantly relocated to a fixed destination cell (*white hole*). The state and action spaces are identical to `OGBench AntMaze`.

- **Teleport topology:** There are two black-hole cells at grid positions $(4, 6)$ and $(5, 1)$, and three white-hole cells at $(1, 7)$, $(6, 1)$, and $(6, 10)$. The deterministic mapping is fixed as $(4, 6) \rightarrow (1, 7)$ and $(5, 1) \rightarrow (6, 10)$.

- **Asymmetry:** Traversing a teleport edge costs 1 step in the black$\rightarrow$white direction, while the reverse path requires navigating around the maze, potentially incurring many more steps. This produces strong asymmetries in the ground-truth MAD that a symmetric metric cannot capture.

- **Dataset:** We load the stochastic `antmaze-teleport-explore-v0` dataset from OGBench and curate it to be consistent with the deterministic mapping: any episode containing a transition through the wrong white hole is discarded entirely. The remaining trajectories reflect only the fixed teleport mapping.

- **Ground-truth MAD:** Computed via BFS on the maze grid with directed teleport edges — the black$\rightarrow$white hop counts as one step, but no reverse edge is added. This directly encodes the asymmetric reachability structure into the ground truth.

## I. Planning Experiments

To assess the practical utility of the learned MAD embeddings, we evaluated the performance of our algorithms and baselines on a downstream goal-reaching task in the OGBench PointMaze and Antmaze environments.

### Planning algorithm

We employed a simple planning algorithm based on random shooting, a form of model-predictive control (MPC), which allows for a direct evaluation of the distance metric as a planning heuristic. This approach isolates the effectiveness of the learned metric from confounding factors that would be introduced by more complex planners.

The planning process at each time step $t$, given a current state $s_t$ and a goal state $g$, is as follows:

1. We generate $K = 100$ candidate action sequences, each of length $H$. The method of generation depends on the environment complexity:

   - **PointMaze:** We sample action sequences uniformly at random on a discretized action space and use the environment simulator to roll out the resulting state trajectories.
   - **AntMaze:** Due to the high-dimensional action space and complex dynamics, random action sampling rarely produces coordinated locomotion. Instead, we select $K$ trajectory segments from the *navigate* dataset. Specifically, we identify the state in the dataset closest to the current state $s_t$ using the learned distance $d_\theta$ and extract K subsequent trajectory segments originating from that neighbourhood.

2. For each of the $K$ action sequences, use the true environment simulator to roll out the corresponding state trajectory $\{s_{t+1}, \ldots, s_{t+H}\}$.

3. Score each trajectory by finding the state within it that minimizes the learned distance to the goal. The score for a trajectory is given by $\min_{0 < i \leq H} d_\theta(s_{t+i}, g)$, where $d_\theta$ is the learned distance.

4. Identify the action sequence that achieved the minimum score (i.e., the one that brought the agent closest to the goal).

5. Execute the first action from this best-scoring sequence to transition to the next state, $s_{t+1}$.

This entire process is repeated at each step in a receding-horizon fashion until the agent reaches the goal or a maximum episode length is exceeded.

Our choice of this simple planning framework is deliberate. By relying on the true simulator and action sampling, the success of the planner depends directly on the metric's ability to provide a meaningful and accurate signal for progress toward the goal. This avoids confounding the evaluation with inaccuracies that might arise from a learned dynamics model or the complexities of a separate policy optimization algorithm.

It is important to note the limitations of this planner: because actions are sampled randomly, the resulting trajectories are sub-optimal and tend to explore only a local region around the agent's current state. Therefore, success in these long-horizon tasks relies heavily on the learned metric providing a consistent and reliable global signal toward the goal, guiding the planner effectively despite its limited local search.

**Evaluation Protocol**

Each task in OGBench accompanies five pre-defined state-goal pairs for evaluation. To ensure statistical robustness, we evaluate over 5 independent random seeds. For each seed and each of the five state-goal pairs, we run 50 evaluation episodes, each with slightly randomized initial and goal states. Performance, as reported in Table 1, is measured by the average success rate across all episodes. An episode is considered successful if the agent reaches a state within a small Euclidean distance of the goal coordinates.

