# OpenReview forum: "Learning the Minimum Action Distance"
_ICML.cc/2026/Conference — ICML 2026 regular_

### Official Review · Reviewer_KS3e · 2026-02-21

**Soundness:** 3
**Presentation:** 3
**Significance:** 3
**Originality:** 3
**Overall Recommendation:** 4
**Confidence:** 3

**Summary:**

The paper proposes an objective function to measure the ‘Minimum Action Distance’ (MAD) in RL. The MAD measures the minimum number of actions that it takes to move from one state to another. Past works have shown this metric to be useful for goal-reaching tasks and reward shaping. The paper proposes two offline algorithms to learn the MAD using only state trajectories. Next, the paper proposes a quasimetric distance function. Finally, the proposed method tested a suite of goal-reaching environments, comparing the learned distance metric to the ground truth and using them for planning.

**Compliance With Llm Reviewing Policy:**

Affirmed.

**Final Justification:**

I find the paper interesting, and the authors have been highly responsive during the rebuttal. I am not assigning a 5 primarily due to the current writing, particularly the insufficient motivation for the loss function. While the authors addressed this and other concerns in the rebuttal, fully incorporating these changes would substantially alter the paper and likely warrant another round of review.

**Key Questions For Authors:**

1. It is stated that the MAD depends solely on the transition function, and not on any policy. I am finding it hard to believe that this is true with the proposed objective. For instance, if states $s_1$ and $s_2$ are one-action away in the MDP, let us assume that all trajectories in the offline-dataset that has these 2 states have at least N>1 actions between them. Could you please explain how the optimization will end up learning $d_\theta(s_1, s_2)= 1$?
2. L:55-56 - “..incorporate both short- and long-term information about how distant two states are from one another.” - What does the short and long term information here mean? This either needs more explanation or is misleading.
3. In L:75, it is stated that SSP is a function of a policy (and the transition function). In L:80, MAD is stated to be a function of just the transition function. But in R:153, it is stated that MAD and SSP are equivalent in deterministic MDPs. Does this equivalence hold for any policy?
4. The TD-version of MAD, TDMadDist, is poorly motivated. Why and when would one want to use this over MadDist?
5. I can not understand the difference between the first and the second stated contributions in the paper: (1) two novel offline algorithms for learning MAD;  (2) a novel quasimetric distance function. What exactly does the second contribution refer to?

---

My current score is based on the raised weakness (specifically the first) and the uncertainties. I will adjust this based on responses to questions.

**Limitations:**

yes

**Strengths And Weaknesses:**

**Strengths**
1. The MAD and need for it is well motivated. Specifically, the comparison of MAD with other methods in Section 2 helps one understand why MAD is better for goal-reaching tasks with asymmetric transition dynamics.
2. The proposed objective is theoretically sound.

---

**Weakness**
1. The MadDist objective contains three individual loss functions, and is poorly motivated and ablated. The paper is missing an important ablation section which should showcase the influence of each of the individual loss functions and the associated hyper-parameters. Specifically, I can not understand what is the role of $L_r$, given that it also serves the same purpose as $L_o$ (Lines A:787, A:798). Similarly, the role of $L_c$ - it is not clear to me why this upper-bound hyper-parameter $H_c$ is used. The paper could greatly benefit from addressing this.
2. The conclusions are drawn based on 3 random seeds, making the claims not strong.
3. The proposed method is not evaluated on teleport environments in OGBench, which is a strong case for an MDP with asymmetric dynamics.
---


**Minor Comments**
1. The paper lists the suite of environments as a contribution, implying they are novel, when most are existing. I'd suggest either removing this or rephrasing to clarify that the contribution is in their ‘curation’ rather than their creation.
2. The last paragraph in Section 1 (L:66-70) feels out-of-place. I would suggest moving it before the contribution paragraph.
3. Section 3 could include a formal definition of “undirected” or “asymmetric” transition dynamics.
4. A:734 - The word ‘minimum’ is enclosed by stars. (typo)


---

- L:10 means line 10, left paragraph
- R:10 means line 10, right paragraph
- A:10 means line 10 in the Appendix

---

> ### Author Rebuttal · Authors · 2026-03-31
>
> **Weakness 1**
>
> The three terms $L_o$, $L_r$, and $L_c$ have distinct roles. $L_o$ is a trajectory-based regression term that uses state pairs $(s_i,s_j)$ sampled from the same trajectory and encourages $d_\theta(s_i,s_j)$ to match the trajectory-defined upper bound $(j-i)$. The scale invariance of this term is important because it ensures that distant states along a trajectory do not dominate the objective purely due to their higher-magnitude errors.
>
> In contrast, $L_r$ is a contrastive loss computed using state pairs randomly sampled from the entire dataset. Its role is not to fit upper bounds on distances using trajectory information, but rather to encourage global separation between unrelated states. Without this term, the learned distance function would be determined primarily by pairs drawn from the same trajectory, and there would be less pressure to separate states that do not co-occur locally in the dataset.
>
> Finally, $L_c$ explicitly enforces the trajectory-based upper bound constraint by penalising cases in which $d_\theta(s_i,s_j) > (j-i)$. This plays a different role from $L_o$: while $L_o$ encourages distances to *approach* the upper bound, $L_c$ prevents them from *exceeding* it. The hyperparameter $H_c$ limits this constraint term to pairs whose index difference is at most $H_c$, rather than enforcing this constraint for all trajectory pairs.
>
> We performed an ablation study analysing the contribution of the $L_r$ across two environments: AntMaze Navigate (expert policies), and Antmaze Explore (random policies).
>
> The attached figures (https://imgur.com/a/dq7HzqW) report the relative change as a function of the weight $w_r$ associated with $L_r$. Across both environments, increasing $w_r$ consistently improves the performance over the baseline $w_r=0$, confirming that $L_r$ provides a complementary learning signal rather than duplicating the role of $L_o$.
>
> In AntMaze Explore, where trajectories are less informative, the gains from $L_r$ are substantially larger and more consistent across distance ranges. In contrast, in AntMaze Navigate, where trajectories already contain structured expert behavior, the improvements remain positive but are more modest. This supports the interpretation that $L_r$ is particularly valuable in low-quality or stochastic data regimes.
>
> We also evaluate the role of the parameter $H_c$ ( https://imgur.com/a/3qWXyAP), the ablation on AntMaze Explore show that increasing $H_c$, consistently improves metrics, particularly at shorter ground-truth distances.
>
> ---
>
> **Weakness 2**
>
> We are currently in the process of running experiments with more seeds across the benchmarks. As an initial step, we have extended the AntMaze Explore experiments from 3 to 5 random seeds (https://imgur.com/a/EL5w8Cc), and the results remain consistent with the original findings.
>
> ---
>
> **Weakness 3**
>
> Deriving the exact ground-truth Minimum Action Distance (MAD) in these environments is non-trivial. Our evaluation protocol relies on environments where the true MAD can be computed exactly, enabling a precise and controlled assessment of representation accuracy.
>
> ---
>
> **Question 1**
>
> The key distinction is between the target quantity being learned and the data available to estimate it. The MAD itself depends only on the support of the transition kernel, and therefore not on any policy. However, when estimated from offline trajectories, the quality of the estimate necessarily depends on dataset coverage, which is inherently induced by a behaviour policy. Thus, in the reviewer's example, if $s_2$ is truly reachable from $s_1$ in one step, but that transition never appears in the dataset, then the exact value $d_{MAD}(s_1,s_2)=1$ is not identifiable from that data alone. This limitation is not specific to the proposed method, but is inherent to learning from offline data generally.
>
> ---
>
> **Question 2**
>
> see Rebuttal reviewer Q7SX.
>
> ---
>
> **Question 3**
>
> No, the equivalence only holds for an optimal goal-reaching policy. In a deterministic MDP, the MAD is equivalent to the optimal stochastic shortest-path distance, because both reduce to the length of the shortest feasible path to the target state. Under a suboptimal policy, the stochastic shortest path may exceed the MAD.
>
> We have clarified the relationship between the MAD and the SSP in the revised version of the manuscript.
>
> ---
>
> **Question 4**
>
> see Rebuttal reviewer Pfcj.
>
> ---
>
> **Question 5**
>
> The first contribution refers to the proposed *learning algorithms* (MadDist and TDMadDist) used to approximate the MAD from offline state trajectories. The second refers to the proposed *quasimetric distance function*, $d_{simple}$, which defines the form of the learned asymmetric distance and shapes the optimisation problem these algorithms solve. Its benefits are that it is computationally efficient, satisfies identity, non-negativity, and triangle inequality, and outperforms existing quasimetric formulations in our ablations.

---

> > ### Author Rebuttal · Reviewer_KS3e · 2026-04-04
> >
> > > Deriving the exact ground-truth Minimum Action Distance (MAD) in these environments is non-trivial. Our evaluation protocol relies on environments where the true MAD can be computed exactly, enabling a precise and controlled assessment of representation accuracy.
> >
> > Given that you report results for AntMaze, why is it not feasible in Teleport? I’m interested in this setting because you note that other methods like the Laplacian representation may be poorly suited to environments with irreversible or asymmetric dynamics (R:101). Including experiments in larger environments with these properties would help clarify whether MAD captures these more challenging settings.

---

> > > ### Author Response · Authors · 2026-04-07
> > >
> > > ***Given that you report results for AntMaze, why is it not feasible in Teleport? I’m interested in this setting because you note that other methods like the Laplacian representation may be poorly suited to environments with irreversible or asymmetric dynamics (R:101). Including experiments in larger environments with these properties would help clarify whether MAD captures these more challenging settings.***
> > >
> > > Our earlier response referred specifically to the original AntMaze Teleport setting, where teleportation is stochastic: each teleport can send the agent to different target locations with uniform probability. In that case, deriving the exact ground-truth MAD with graph-based shortest-path methods is non-trivial, which is why we did not include it in the original analysis.
> > >
> > > To address the reviewer’s concern, we considered a deterministic but still strongly asymmetric variant with deterministic teleportation, where each teleport always maps to a fixed target location. In this setting, we can compute the exact ground-truth MAD on the discretized grid using Floyd–Warshall, and we report the resulting Pearson correlation, Spearman correlation, and Ratio CV.
> > >
> > > As shown in the attached results ( https://imgur.com/a/l3L7FDh ), MadDist achieves the best overall performance, with TDMadDist consistently second. Both methods substantially outperform Hilbert, which is unable to represent asymmetric distances, and also improve over QRL. These results support our claim that MAD is well suited to large/continous environments with irreversible or asymmetric dynamics.
> > >
> > > MadDist exhibits a slightly larger RatioCV than TDMadDist. The RatioCV metric measures how well distortions in the learned distances can be explained by a simple constant rescaling of the ground-truth distances; lower values therefore indicate that the learned distances are closer to a uniform global scaling.
> > >
> > > This behavior is related to the additional contrastive term in MadDist, which globally pushes all state pairs.
> > > This global repulsive force encourages separation across the state space rather than preserving a single proportional scaling of distances. As a result, the learned distances may deviate slightly from a uniform rescaling of the ground-truth values, leading to a modestly higher RatioCV. In contrast, the TD formulation tends to propagate distances through local bootstrapping updates, which can lead to a more uniform scaling behavior.
> > >
> > > Importantly, MadDist achieve strong correlation with the ground-truth distances, indicating that the learned geometry is accurate in terms of structural fidelity, which is the primary objective in our setting.
> > >
> > > We agree that this is an important regime, and we will add this discussion and these results in the revised version.
> > >
> > > ---
> > >
> > > We thank the reviewer for the constructive feedback and engagement during the rebuttal process. We hope our responses clarify the remaining questions, and we will incorporate the new results and suggested improvements in the final version of the paper.

---

### Official Review · Reviewer_Q7SX · 2026-03-12

**Soundness:** 2
**Presentation:** 3
**Significance:** 2
**Originality:** 2
**Overall Recommendation:** 2
**Confidence:** 3

**Summary:**

The work explores a deep learning based approach to estimate the minimum action distance (MAD) for any given state pairs. The authors formulate the problem as a constraint optimization problem and then use a neural network to learn a kernel that can be used with a distance function to estimate MAD values. The primary contribution is development of two algorithms that can be used to estimate the kernel function for calculating MAD values.

**Compliance With Llm Reviewing Policy:**

Affirmed.

**Final Justification:**

I think the authors have cleared most of my concerns. However, there were multiple places in the main text where the exact contribution was not clear. I think the work holds value but needs work on its presentation. As such, I would like to maintain my score as the manuscript requires major revisions.

I think revisions along the following points will make it clearer:
- The contributions need to be clearer: The theoretical construction itself is a contribution. The scalability part needs to be rephrased and the caveats for classical methods need to be elaborated on (or the scope of the work needs to be reduced).
- I agree that the use of the word "quasimetric" depends on the definition and context. However, it needs more elaboration for a general audience (part of which can be in the main text and rest in appendix). One of the baselines used have different definitions of quasimetric and this needs to be discussed.
- The authors need to be clear about their construction for the continuous space formulation. While the change that the authors mentioned during rebuttal was obvious to them, it is not for the reader. Furthermore, some evaluations are on continuous spaces and experiments showing the effect of different $\Delta t$ values will show the robustness of the formulation.

While major part of the revision is in presentation, I think it needs to go through another round of peer review.

**Key Questions For Authors:**

1. In the introduction, the authors claim that their method incorporates both short term and long term information about distant states. Can the authors clarify what they mean by "incorporating both short term and long term information" as compared to other baselines or methods presented? Specifically, is this unique to the presented method?

2. For the continuous case, while the distance function is defined, it is not clear how $d_{MAD}$ is defined. Can the authors explain how $d_{MAD}$ is finite for pairs in $(s, s') \in \mathcal{S}^2 / \mathcal{R}$?

3. For the scalability argument for learning based approach in comparison to classical flow based method, it is not clear why they are better. While I agree that for continuous space classical flow based method may require some non-trivial method for discretization, for discrete space that doesn't seem to be the case. Authors mention Floyd-Warshall algorithm which has complexity of $|\mathcal{S}|^3$. However, algorithms like Johnson's algorithm have a complexity of $|V|^2 \log |V| + |V||E|$ where $V$ would be the set of states in the training dataset and $E$ the transitions between them. The presented algorithms calculate an expectation over state pairs in the dataset. The training would require to go over all possible pairs in a given trajectory. Given the added compute from the neural network as well as the need to repeat the process multiple times for convergence, it does not appear trivial to see the benefit. Can the authors provide an analysis (and maybe add it in the appendix)?

**Limitations:**

yes

**Strengths And Weaknesses:**

**Strengths:**
1. The authors pose the problem as a constraint optimization problem. Posing it as an optimization problem gives a clear idea of how classical flow optimization based approaches can be used to approach the problem of estimating MAD values.

2. Overall, the entire paper is well written to explain the concepts to the reader.

3. The authors do a comprehensive set of experiments to evaluate their method. The experiments cover both discrete and continuous observation and action spaces. The experimental setup and evaluation is well explained and well motivated.

**Weaknesses**:

1. The authors claim that the constraint optimization problem can be extended to continuous domain without providing the actual formulation. The shift from discrete to continuous version does not seem to be trivial.

2. It is not clear why the presented method is more scalable to other methods using neural networks. From what I understand, the authors are trying to compare the complexity of the classical flow based approach to a learning based approach. However, it is not clear if that is the case. I reached the conclusion based on no scalability comparisons with other learning based methods and not because it is explicitly mentioned. Furthermore, there is no analysis of why learning based methods are more scalable in terms of actual computation. (Refer to my questions on the same to see what I mean by analysis)

3. The definition given for quasimetrics seems to be missing the separation axiom $d(x,y) = 0 \implies x = y$. The authors seem to have defined the Lawvere metric which drops the separation axiom in addition to symmetry $d(x,y) = d(y,x)$. The $d_{simple}$ quasimetric presented by the authors is also not a quasimetric as it is possible to have $d(x,y) = 0 \text{ and } x \neq y$. Example: $x = [0, 0], y = [1, 1]$

---

> ### Author Rebuttal · Authors · 2026-03-31
>
> **Question 1**
>
> We thank the reviewer for this question and are happy to clarify what we mean by incorporating both short- and long-term information.
>
> What we intended to convey is that our objective uses information from state pairs separated by different numbers of transitions within the same trajectory. Specifically, the formulation enforces constraints derived from one-step transitions, which capture local structure, while also using trajectory index differences $(j-i)$ for pairs of states that may be many steps apart. As a result, the learning signal is defined not only for consecutive states, but also for distant states along the same trajectory.
>
> For any trajectory $\tau = (s_0, \ldots, s_n)$, the quantity $(j-i)$ provides a valid upper bound on the distance between $s_i$ and $s_j$, because the trajectory itself provides a path of that length. We use these bounds for multiple values of $j-i$, thereby incorporating supervision from both short separations (e.g., $j-i=1$) and longer separations (e.g., $j-i \gg 1$).
>
> This property is not entirely unique to our method, but it is implemented differently from several baselines. In particular, some prior approaches rely primarily on local transition constraints or pairwise comparisons, whereas our formulation explicitly incorporates distance signals between arbitrary pairs of states along a trajectory, enabling information to propagate more efficiently over long horizons.
>
> We have updated the wording in the revised manuscript to make this point more precise.
>
> ---
>
> **Question 2**
>
> Our formulation indeed assumes discrete, uniform decision steps, which is the standard setting for Markov Decision Processes. Specifically, the dynamics are modelled as a time-homogeneous discrete-time process indexed by $t = 0, 1, 2, ...,$ where each step corresponds to one application of the transition kernel $P(\cdot | s, a)$. The Minimum Action Distance is therefore defined as the minimal number of such transitions required to reach a target state.
>
> Importantly, this assumption concerns the time index, not the state space: the state space $S$ may be continuous, but the system evolves over a series of discrete decision steps. The resulting distance is thus measured in units of transitions rather than physical time.
>
> If one were to consider variable-duration transitions or continuous-time dynamics ( e.g., semi-Markov or continuous-time MDPs), the natural generalization would be to replace the step count with a cost or elapsed-time function. Our current work focuses on the standard discrete-time setting.
>
> Formally, the reachability relation is defined via the support of the transition kernel:
>
> $$ R = \{(s, s') \in S^2: \exists a\in A \text{ such that } P(s'|s, a) > 0\}. $$
>
> This definition remains valid for continuous state spaces because it depends only on the transition support, not on the enumeration of states.
>
> Finiteness of $d_{MAD}(s, s')$ then follows from the existence of a finite-length trajectory in discrete time. If there exists a sequence
>
> $$ s=s_0 \rightarrow s_1 \rightarrow ... \rightarrow s_k = s' \text{ with } (s_i, s_{i+1}) \in R,$$
>
> then by definition
>
> $$d_{MAD}(s, s') \leq k < \infty.$$
>
> We will clarify this modelling assumption explicitly in the revised version.
>
> ---
>
> **Question 3**
>
> We agree that in purely discrete settings with a fully known transition graph, classical algorithms such as Johnson’s can be more efficient than Floyd-Warshall, and our intent was not to claim asymptotic superiority over these methods.
>
> The key distinction we aim to highlight is the regime where the state space is continuous or very large, making it impractical to explicitly construct the transition graph. In such cases, classical shortest-path methods require enumerating states and transitions, whereas our approach learns a parametric distance function directly from sampled trajectories without constructing the full graph.
>
> Regarding computational cost, our training does not enumerate all state pairs; instead, it operates on sampled minibatches, so the per-iteration complexity scales with batch size. The main advantage of the learning-based approach is therefore amortization and generalization: once trained, the model provides constant-time distance evaluation and can estimate distances for previously unseen states.
>
> ---
>
> **Weakness 3**
>
> We thank the reviewer for pointing this out.
>
> We agree that our definition does not enforce the separation axiom $d(x,y) = 0 \implies x = y$; formally, our distance is closer to a quasi-pseudometric (or Lawvere metric) than to a strict quasimetric. This distinction is terminological and does not affect our results, because the optimization and guarantees rely only on non-negativity and the triangle inequality. We have clarified the definition and updated terminology used in the revised version of the manuscript.

---

> > ### Author Rebuttal · Reviewer_Q7SX · 2026-04-03
> >
> > Continuous Space Formulation:  The dependency of MAD on the sampling frequency of a continuous trajectory weakens its use as a lower bound. Just reducing the sampling frequency by half also reduces the maximum MAD values by half. I understand that the data is in a dataset and a standardized dataset or simulation data can have the same sampling frequency but this is hard to ensure for collected data. Furthermore, for a dataset collected at a high sampling frequency will have huge upper bounds which can make MAD values as a lower bound vacuous. If the reviewers can provide some empirical data or an argument why that will not be the case, it will be helpful.
> >
> > Scalability Claim: I agree with the authors that neural approaches help train a kernel function which can be used to map continuous states or similar discrete states to a same (or close by vector). Can such a compression not be achieved using something like a VAE? Can the authors explain how their method is more scalable than other neural approaches used as baselines?
> >
> > Metric Definition: If the metric does not need to have the separation axiom, can the authors clarify why previous works required this? Is there a tradeoff in terms of convergence guarantees? In general, can the authors give more details about how this changes their work in comparison to previous works?

---

> > > ### Author Response · Authors · 2026-04-07
> > >
> > > ***Continuous Space Formulation***
> > >
> > > Our formulation explicitly enforces the one-step constraint $d(s, s') \le 1$ for all $(s, s') \in R$, which defines the unit of distance. If the time between decision steps is $\Delta t$ rather than a normalized unit step, the constraint can be written equivalently as $d(s, s') \le \Delta t$, and all distances scale accordingly. In other words, the constant $1$ in the constraint represents a choice of units rather than a structural assumption.
> > >
> > > This formulation also naturally extends to variable sampling frequencies. If transitions/decisions occur at non-uniform time intervals $\Delta t_i$, the corresponding constraint can be written as $d(s_i, s_{i+1}) \le \Delta t_i$, and distances accumulate according to the sum of these intervals. In this case, the distance reflects elapsed decision time rather than the number of steps, making the formulation robust to irregular sampling.
> > >
> > > Moreover, our evaluation metrics such as Spearman or Pearson correlation depend on relative structure rather than absolute scale and are therefore invariant to uniform rescaling of distances. This reflects our primary goal: ensuring that states are ordered correctly by distance, rather than recovering a particular numerical scale.
> > >
> > > ***Scalability Claim***
> > >
> > > In principle, one could train a Vector Quantized VAE to obtain a discrete latent representation and then apply graph algorithms such as Floyd–Warshall. However, this requires a discretization that preserves reachability structure. A standard VAE is trained using reconstruction loss, which groups states by visual similarity rather than by transition dynamics or control distance.
> > >
> > > As a result, we do not know how to discretize the state space so that nearby latent states correspond to states reachable in few steps. Such a discretization would require additional supervision aligned with dynamics.
> > >
> > > Moreover, one cannot generally assume that nearby states along a trajectory are close in control distance and that distant states are far, without strong assumptions about the behaviour policy that generated the data (e.g., in the presence of looping or exploratory trajectories).
> > >
> > > Our constrained optimization formulation circumvents this issue by relying only on valid upper-bound constraints derived from observed transitions, together with the property that the Minimum Action Distance (MAD) is the unique maximal feasible solution of our constrained optimization problem. This ensures correctness without requiring heuristic discretization or policy-dependent assumptions, and without requiring a reconstruction objective allowing us to approximate the relevant control distance in a more principled and efficient way.
> > >
> > > In our experiments, we compare against neural baselines such as QRL and Hilbert, which also aim to learn the minimum action distance function directly from data. Our empirical results show that our formulation consistently outperforms these methods in terms of accuracy in approximating the target distance. Moreover, these improvements in distance accuracy translate into better performance in downstream planning tasks.
> > >
> > > To clarify, our claim is not that our method is inherently more scalable than other neural approaches, but rather that it provides a more faithful approximation of the desired distance under the same learning setting.
> > >
> > > ***Metric Definition***
> > >
> > > The separation axiom is not needed for our optimization problem or theoretical guarantees. Our analysis only relies on $d(x,x)=0$, non-negativity, and the triangle inequality. These are sufficient to prove that feasible solutions are upper-bounded by $d_{\text{MAD}}$ and that the objective recovers the desired distance. Thus, allowing $d(x,y)=0$ for some $x \neq y$ does not affect correctness. (see appendix A and C)
> > >
> > > There is therefore no trade-off in convergence guarantees. The correctness of our method depends on feasibility and optimality of the constrained problem, not on explicitly enforcing separation. In practice, the optimal solution satisfies separation wherever the true minimum action distance is positive, while allowing more flexibility during optimization.
> > >
> > > More broadly, this is not specific to our construction: the same observation applies to prior quasimetric parameterizations such as IQE ([1], [2]) that we use in our comparison, which also do not enforce the separation axiom in general. For example, taking $x=[1, 1]$ and $y = [0, 0]$, each IQE interval becomes $[x_j, x_j]$ which has zero Lebesgue measure; hence $d_{IQE}(x, y) = 0$ despite $x \neq y$. The main difference is therefore not in convergence behaviour, but in terminology. We will clarify in the revised version that the relevant object for our analysis is more precisely a quasi-pseudometric/Lawvere-style distance, and that explicit separation is unnecessary.
> > >
> > > [1] Optimal Goal-Reaching Reinforcement Learning via Quasimetric Learning
> > >
> > > [2] Improved Representation of Asymmetrical Distances with Interval Quasimetric Embeddings

---

### Official Review · Reviewer_29TV · 2026-03-12

**Soundness:** 2
**Presentation:** 3
**Significance:** 2
**Originality:** 2
**Overall Recommendation:** 3
**Confidence:** 4

**Summary:**

This work proposes a method for learning minimum action distance, meaning a distance metric that given two states $s_1$ and $s_2$, captures the minimum number of action steps needed to go from $s_1$ and $s_2$ in the given MDP. Notably, the proposed method works when the distance is asymmetric. The authors then test the proposed method on distance learning on such environments as point maze, ant maze, key door and cliff walk.

**Compliance With Llm Reviewing Policy:**

Affirmed.

**Final Justification:**

The experiments partially addressed my concerns, hence i raised my score. However, the original paper does seem to be a bit too raw to warrant acceptance.

**Key Questions For Authors:**

1. Could you provide results for [1] on the benchmarks you used?
2. Can you highlight what the exact differences between MadDist and [1] are?
3. Is TDMadDist stricly worse than MadDist throughout?

**Limitations:**

Authors haven't discussed limitations, which I believe include limited evaluation, and unclear applicability to more complex environments.

**Strengths And Weaknesses:**

**Strengths:**
- Results are strong when compared to Hilbert representations and QRL
- Method is quite clear and simple

**Weaknesses:**
- The method is only evaluated on state-based environments. To strengthen the claim, would be great to demonstrate that it works with image-based state spaces.
- According to the presented results, TDMadDist is not any better than MadDist. It is then unclear why it was proposed at all.
- The objective proposed is quite similar to prior work of Steccanella and Jonsson [1], with minor differences in denominators, making the method contribution less significant. Additionally, the authors do not compare to the method proposed in [1]. I believe this comparison is essential to justify MadDist.

[1] Steccanella, L. and Jonsson, A. State Representation Learn-
ing for Goal-Conditioned Reinforcement Learning. In
Joint European Conference on Machine Learning and
Knowledge Discovery in Databases, pp. 84–99. Springer,
2022.

---

> ### Author Rebuttal · Authors · 2026-03-31
>
> **Weakness 1**
>
> While our current evaluation focuses on vector-based state environments due to computational constraints, we would like to emphasize that some of our benchmarks, particularly AntMaze, already operate in highly complex, high-dimensional settings. This environment combines long-horizon navigation with challenging locomotion dynamics.
>
> Concretely:
>
> - State space: The agent observes a 29-dimensional vector.
> - Action space: An 8-dimensional continuous control signal corresponding to joint torques.
>
> These characteristics make AntMaze a demanding testbed that captures many of the challenges present in real-world continuous-control problems, even though observations are not pixel-based.
>
> More importantly, our method is largely agnostic to the observation modality, as it operates on learned state representations rather than raw inputs. In principle, the same objective could be applied on top of visual encoders, as is common in representation learning pipelines.
>
> ---
>
> **Weakness 2**
>
> The goal of TDMadDist is not to outperform MadDist, but to show that the MAD objective can be formulated within a TD framework while remaining consistent with the underlying constrained optimization problem. This provides a closer conceptual comparison to TD-based approaches such as Hilbert.
>
> Empirically, while TDMadDist underperforms MadDist, it still outperforms TD-based baselines (e.g., Hilbert), indicating that the proposed constraints and parameterization remain beneficial even in a bootstrapped setting.
>
> We will clarify its role as an exploratory extension, with future work needed to assess its potential advantages over MadDist (e.g., in online settings).
>
> ---
>
> **Question 1**
>
> While our objective is indeed inspired by Steccanella and Jonsson [1], the difference is not merely a minor modification of the loss. Their formulation fundamentally relies on a symmetric distance function, which introduces a well-known limitation: it cannot represent the inherently asymmetric nature of the true MAD. As a result, their method effectively approximates a symmetrized version of the distance, i.e., it collapses directional structure and underestimates the true MAD in asymmetric environments (e.g., capturing $\min(d_{MAD}(s_i, s_j), d_{MAD}(s_j, s_i)) \forall (s_i, s_j) \in \mathcal{S}^2$
>
> This limitation is explicitly acknowledged in their work and is shared by other approaches based on symmetric metrics. In contrast, our method is specifically designed to overcome this issue by using a quasimetric formulation, which allows us to model directional distances and recover the true MAD in asymmetric settings.
>
> For this reason, we consider comparisons against methods that already address asymmetry (e.g., quasimetric-based approaches) to be more informative and aligned with the core contribution of our work. Including [1] would primarily highlight a known limitation rather than provide additional insight into the effectiveness of our approach.
>
> That said, we agree that including this comparison could help make this distinction more explicit. We will make sure to add results or a discussion in the final version to clarify the impact of symmetry on performance.
>
> **Question 2**
>
> The differences between MadDist and Steccanella and Jonsson [1] go beyond minor variations in the objective and affect both the formulation and the resulting behavior:
>
> - Asymmetric distance (quasimetric):
> Our method replaces the symmetric metric used in [1] with a quasimetric, allowing us to model inherently asymmetric distances. This is a fundamental distinction, as it enables recovery of the true MAD in environments with directional dynamics, which symmetric formulations cannot capture.
>
> - Additional contrastive term ( $L_r$ ):
> We introduce a second loss term that samples arbitrary state pairs across the dataset, not just pairs from the same trajectory. This encourages global separation between unrelated states and provides a stronger, more informative learning signal beyond trajectory-local supervision.
>
> - Scale-invariant (normalized) objective:
> Our objective is scale-invariant, avoiding the bias of squared losses in [1] that overemphasize long-horizon pairs.
>
> ---
>
> **Question 3**
>
> Across our experiments, TDMadDist consistently underperformed MadDist.
>
> However, the goal of TDMadDist is not to outperform MadDist, but to demonstrate that the MAD objective can be formulated within a TD framework while remaining consistent with the underlying constrained optimization problem. This enables a more direct comparison to TD-based approaches such as Hilbert.
>
> Empirically, while TDMadDist underperforms MadDist, it still outperforms TD-based baselines (e.g., Hilbert), indicating that the proposed constraints and parameterization remain beneficial even in a bootstrapped setting.
>
> We will clarify its role as an exploratory extension, with future work needed to assess its potential advantages over MadDist (e.g., in online settings).

---

> > ### Author Rebuttal · Reviewer_29TV · 2026-04-01
> >
> > Thank you for your response.
> > You're right, I understand that comparison to [1] is not entirely apples to apples as they learn a symmetric distance. What about plugging in an asymmetric distance as you showed in the paper, like $d_\mathrm{simple}$, or $d_{WM}$ in the method proposed in [1]? That connects to Question 2 about the differences. Would that be a reasonable ablation?
> >
> > Regarding TD version, are there any settings where TD would be preferable to non-TD?

---

> > > ### Author Response · Authors · 2026-04-07
> > >
> > > ***Thank you for your response. You're right, I understand that comparison to [1] is not entirely apples to apples as they learn a symmetric distance. What about plugging in an asymmetric distance as you showed in the paper, like $d_{simple}$, or $d_{WN}$ in the method proposed in [1]? That connects to Question 2 about the differences. Would that be a reasonable ablation?***
> > >
> > > We thank the reviewer for this thoughtful suggestion. To clarify this point, we performed additional experiments ( https://imgur.com/a/UMb884v ) using the objective of [1] with two distance parameterizations: the original symmetric $L_1$ distance (denoted [1]) and the same objective combined with our asymmetric $d_{\text{simple}}$ parameterization (denoted [1] – $d_{\text{simple}}$).
> > >
> > > We evaluate these variants on CliffWalking, which exhibits asymmetric dynamics, and AntMaze Explore, which is symmetric.
> > >
> > > On CliffWalking, both MadDist and TDMadDist achieve the best performance across metrics. As expected, the original formulation [1], which is intrinsically symmetric, performs worst. Replacing its distance with the asymmetric $d_{\text{simple}}$ substantially improves results, but it still remains clearly below MadDist and TDMadDist. This indicates that the improvement does not stem solely from the distance parameterization; the optimization objective itself plays a critical role.
> > >
> > > In contrast, on AntMaze Explore, MadDist and TDMadDist again outperform both variants of [1]. However, in this symmetric setting, the original symmetric version [1] performs better than [1] – $d_{\text{simple}}$, which is consistent with the symmetric inductive bias being beneficial when the environment dynamics are symmetric.
> > >
> > > Overall, these results suggest that both components matter: using an asymmetric distance improves performance in asymmetric environments, but the gains of MadDist cannot be explained solely by replacing the distance function within [1].
> > >
> > > ---
> > >
> > > ***Regarding TD version, are there any settings where TD would be preferable to non-TD?***
> > >
> > > We thank the reviewer for the opportunity to further clarify this point.
> > >
> > > In the offline setting considered in this work, the direct MadDist formulation is generally preferable, as it provides a more stable learning signal and achieves better empirical performance. However, TD-based formulations can be advantageous in online settings where a full trajectory dataset is not readily available.
> > >
> > > In particular, TD-style updates are naturally suited to online or streaming settings, where data arrives sequentially and distances must be updated incrementally without revisiting the full dataset.
> > >
> > > Our goal in introducing TDMadDist was to establish that the MAD objective can be expressed within a TD framework, enabling compatibility with these regimes. A full investigation of this would require addressing several design dimensions, such as exploration strategies and online data collection, each of which introduces additional algorithmic and experimental complexity. These questions are orthogonal to the central objective of the present work, which is to study the constrained optimization formulation of MAD and evaluate its effectiveness in the offline representation-learning setting.
> > >
> > > ---
> > >
> > > We thank the reviewer for the constructive feedback and engagement during the rebuttal process. We hope our responses clarify the remaining questions, and we will incorporate the new results and suggested improvements in the final version of the paper.

---

### Official Review · Reviewer_Pfcj · 2026-03-13

**Soundness:** 2
**Presentation:** 3
**Significance:** 2
**Originality:** 3
**Overall Recommendation:** 2
**Confidence:** 4

**Summary:**

This paper presents an offline state representation framework aimed at learning the Minimum Action Distance (MAD) between states without relying on reward signals or action labels. The authors construct a quasimetric embedding space to capture the asymmetric and directed nature of transitions. They propose two self-supervised algorithms (MadDist and TDMadDist) which utilize upper bounds derived from trajectory indices to approximate the MAD. The representations are evaluated on diverse maze environments, showing improved correlation with ground-truth distances compared to symmetric Euclidean baselines.

**Compliance With Llm Reviewing Policy:**

Affirmed.

**Key Questions For Authors:**

1. How do you guarantee convergence (or prevent severe distortion) when offline dataset is populated by a suboptimal behavioral policy where $j-i$ can be way larger than the true MAD?
2. Why was a simplistic random-shooting planner used for high-dimensional continuous control evaluations instead of extracting a standard goal-conditioned policy via an actor-critic method?

**Limitations:**

No. The authors provide a boilerplate Impact Statement but entirely fail to discuss algorithmic limitations. Constructive suggestion: Include a formal analysis on how dataset optimality impacts the tightness of the learned bound.

**Strengths And Weaknesses:**

### Strengths
* **Asymmetric Metric Formulation:** The shift from symmetric distance embeddings (like Hilbert space embeddings) to asymmetric quasimetrics is necessary for capturing directed environmental dynamics.
* **Scale-Invariant Objective:** The use of a scale-invariant loss function prevents long-horizon state pairs from dominating the gradient updates. This balances short-range and long-range structural learning.

### Weaknesses
* **Trajectory Bound Assumption:** The core optimization relies on the assumption that the index difference $j-i$ on a trajectory acts as a tight upper bound for the true MAD. But, $j-i$ is a very loose overestimate in highly stochastic environments or offline datasets collected by a random behavioral policy. This systematically incentivizes the network to overestimate the true MAD.
* **TDMadDist Instability:** The TDMadDist algorithm enforces a hard minimum over a non-stationary bootstrapped target and the trajectory upper bound (Equation 8). Bootstrapping over initially overestimated bounds propagates errors, which explains why TDMadDist systematically underperforms the direct MadDist algorithm in complex environments.
* **Naive Downstream Evaluation:** For continuous control tasks like AntMaze, authors evaluate representations using a primitive random-shooting Model Predictive Control (MPC) planner. This combines the true quality of the learned representation with the extreme inefficiencies of random action sampling in high-dimensional spaces, making it difficult to assess the metric's practical utility compared to standard actor-critic policy extraction.

---

> ### Author Rebuttal · Authors · 2026-03-31
>
> **Weakness 1.**
>
> Our objective is to solve the constrained optimization problem introduced in Eq. (1) and further detailed in Appendices A and C. As shown in Appendix A, this problem admits a unique solution corresponding exactly to the Minimum Action Distance (MAD).
> In particular, any function that maximizes pairwise distances while satisfying the constraints:
>
> - $d(s, s') \leq 1$ for one-step transitions, and
> - $d(s, s') \leq d(s, s'') + d(s'', s')$, the triangle inequality.
>
> is upper-bounded by the true $d_{MAD}$, and the optimal solution recovers it exactly.
>
> Importantly, the index difference $(j - i)$ along a trajectory \textbf{is not assumed to be a tight estimate} of the true MAD. Rather, it provides a valid but potentially loose upper bound, since any observed trajectory induces a feasible path between states. We fully agree with the reviewer that in highly stochastic environments or datasets collected with random policies, this bound can be loose.
>
> However, this does not systematically bias the model toward overestimation. The reason is that the trajectory-based signal is only one component of the learning objective. Specifically:
>
> - The constraint loss enforce $d_\theta(s_i, s_j) <= (j-i)$, preventing violations of the upper bound.
>
> - The contrastive term $L_r$ actively pushes apart unrelated state pairs, encouraging global consistency and preventing trivial or uniformly inflated solutions.
>
> - More fundamentally, the optimization is anchored by the structure of the constrained problem and the tight bound $d(s, s') \leq 1$: any feasible solution must remain below the true MAD, and the objective encourages distances to grow \textbf{only up to the maximum allowed by the constraints}, not beyond.
>
> Therefore, the trajectory index is best understood as a \textbf{supervisory signal that improves sample efficiency when informative}, rather than a defining assumption of the method. Even when this signal is weak, the contrastive objective and the quasimetric structure still guide learning toward a consistent approximation of the MAD.
>
> The current presentation may give the impression that the trajectory-based term plays a dominant role. We will revise the text to clarify the complementary roles of both loss components.
>
> ---
>
> **Weakness 2.**
>
> We agree that bootstrapping (especially early in training) can propagate errors when initial estimates are inaccurate. This is a known limitation of TD-based methods and explains the weaker performance of TDMadDist in more complex settings.
>
> That said, the goal of TDMadDist is not to outperform MadDist, but to show that the MAD objective can be formulated within a TD framework while remaining consistent with the underlying constrained optimization problem. This provides a closer conceptual comparison to TD-based approaches such as Hilbert.
>
> Empirically, while TDMadDist underperforms MadDist, it still outperforms TD-based baselines (e.g., Hilbert), indicating that the proposed constraints and parameterization remain beneficial even in a bootstrapped setting.
>
> We will clarify its role as an exploratory extension, with future work needed to assess its potential advantages over MadDist (e.g., in online settings).
>
> ---
>
> **Question 2.**
>
> The use of a simple random-shooting MPC planner is intentional, as our goal is to isolate representation quality from the confounding effects of policy learning.
>
> Most prior work evaluates representations within RL pipelines, where performance reflects both the metric and policy learning details. For instance, QRL augments DDPG with behavioral cloning, making it hard to attribute gains to the representation alone.
>
> In contrast, our setup explicitly removes these confounds. The planner is deliberately simple so that performance is primarily driven by the learned distance, not by sophisticated action optimization. Introducing actor-critic methods would reintroduce significant variance and obscure the contribution of the representation itself.
>
> Moreover, our evaluation is grounded in environments where the true MAD is known, enabling direct and rigorous assessment of distance accuracy, something not possible in prior benchmarks.
>
> Finally, the downstream planning results should be interpreted as a controlled validation: improvements in the learned metric translate into better goal-reaching performance even under a weak planner.
>
> We will clarify this design choice in the revision to avoid the impression that the planner is meant to be competitive rather than diagnostic.
>
> ---
>
> **Question 1.**
>
> For the analytical part of this question, we refer the reviewer to our response to Weakness 1 above..
>
> Empirically, many of our datasets are collected using purely random policies. A representative example is the AntMaze-Explore setting, where trajectories are generated without directed exploration. Despite this highly suboptimal data collection regime, our algorithm still learns accurate distance representations and achieves strong downstream performance.

---

### Decision · Program_Chairs · 2026-04-30

**Decision:**

Accept (regular)

**Comment:**

The authors propose methods for learning asymmetric distance measures between states to capture underlying MDP dynamics and support downstream RL tasks.  The authors provide insightful and innovative technical contributions and analysis.  They also design clever experiments with well-motivated evaluation metrics to evaluate the performance of their methodology.  With its elucidating empirical analysis, this paper makes very solid scientific contributions and is thus a strong candidate for acceptance at ICML.  Reviewer discussion on this paper was extensive and uncovered many points of clarification and yielded new experimental results that should be incorporated into the paper (or into the appendices and then briefly referenced in the main paper).  I think there were a few reviewer-specific misunderstandings around continuous time models that do not necessarily need to be addressed on revision.  That said, I believe that the clarifications and additional experiments should not be hard to incorporate and will improve on an already solid set of contributions.  For these reasons, I firmly believe this paper passes the acceptance threshold for ICML.